# Climate Change Accounting and Reporting: A Systematic Literature Review

**Carmela Gulluscio [1],\*, Pina Puntillo [2], Valerio Luciani [3] and Donald Huisingh [4]**

1   Department of Law and Economic Sciences, Unitelma Sapienza University of Rome, 00161 Rome, Italy
2   Department of Business Administration and Law, University of Calabria, 87036 Rende, Italy; pina.puntillo@unical.it
3   Department of Legal and Economic Sciences, Unitelma Sapienza University of Rome, 00161 Rome, Italy; valerio.luciani@unitelmasapienza.it
4   University of Tennessee and Bioeconomy in Transition Research Group (BIT-RG) in Unitelma Sapienza University of Rome, I-00161 Rome, Italy; dhuisingh@utk.edu
\*   Correspondence: carmela.gulluscio@unitelmasapienza.it; Tel.: +39-333-170-9372

**Abstract:** During the last few years, sustainability has become an increasingly important dimension for corporations. Many stakeholders expect companies to implement sustainability-oriented practices and report on these actions and their results. As a consequence, corporate accountability and, more specifically, corporate accounting and reporting, should focus not only on financial, social, and environmental performance, but also on sustainability-related aspects. Among these aspects, climate change is becoming increasingly important for companies, which must take action to counter the effects of their activities on climate change and inform their stakeholders about these actions and their effects. Given the initial state of research about climate change accounting and reporting, the authors focused on the sustainable development goal (SDG) no. 13, "climate action", in order to highlight the current state and the future directions of this area of inquiry. They used a mixed approach to perform a systematic literature review about sustainability accounting/reporting and climate change: (1) a qualitative analysis according to a qualitative analytical framework, and (2) a bibliometric (descriptive statistical) approach. The authors found that: (1) the main perspectives addressed in the selected articles relate to sustainability accounting and reporting in a broad sense; (2) there was a lack of contributions about management of climate change-related aspects, with specific reference to strategic and operational planning, accounting, and control of the actions implemented by the management of firms to counter climate change problems. The authors suggested the topics accounting scholars should focus their future research upon and underscored the social responsibilities of accounting scholars to increasingly integrate climate change mitigation into their accounting foci. They reviewed the main areas of climate change accounting/reporting literature and identified the gaps to be filled.

**Keywords:** climate change; sustainability accounting; sustainability reporting; sustainability accountability; sustainable development goals (SDGs); climate action; literature review

---

## 1. Introduction

During the last decade, sustainability has produced increasingly important dimensions for corporations, in part because many of their stakeholders are demanding that they take them into account. Some stakeholders are insisting that companies have a clear and broad sustainability vision and related policies for implementation of sustainability concepts within their corporate governance and business strategies. As a consequence, corporate accountability and, more specifically, corporate accounting, reporting, and assurance activities, should focus not only on financial, social, and environmental

performance, but also on sustainability-related aspects. In this context, a wide range of sustainability-related accounting and reporting practices are emerging.

The authors of this paper refer to the different types of nontraditional forms of accounting and reporting as "sustainability accounting". It is essential to underscore that there are different interpretations of the term "sustainability accounting". As stated by Schaltegger and Burritt [1], it can be understood as:

(1) An illusion and a buzzword;
(2) A concept with multiple meanings, including environmental, social, eco-efficiency, etc., issues;
(3) A broad concept including information about corporate sustainability measurement and management;
(4) A stakeholder engagement process for the development of measurement and management tools related to the economic, social, and environmental aspects and their mutual links.

Recent academic research on accounting and accountability has been oriented towards these issues, placing great emphasis on corporate responsibility and on related forms of nonfinancial disclosure.

The objective of the authors of this paper was to contribute to the special issue topic of "Accounting and Accountability for SDGs" (https://www.un.org/sustainabledevelopment/sustainable-development-goals/). These goals aim at achieving a better and more sustainable future for all and should be achieved by 2030. Among the 17 sustainable development goals (SDGs), the authors focused upon goal # 13, "climate action", aiming to "take urgent action to combat climate change and its impacts" (https://www.un.org/sustainabledevelopment/climate-change/). To this end, this paper focused upon two particular aspects within the special issue topic of "Accounting and Accountability for SDGs": accounting and reporting related to climate change aspects. These aspects could be simply referred to as "accounting and reporting for climate change" [2], "climate change accounting and reporting", or "climate change-based accounting and reporting". According to SDG no. 13, climate change is induced by global warming, which is strongly related to rising $CO_2$ and greenhouse gas (GHG) emissions. To limit climate change, it is important to act on its causes. The interest of accounting scholars towards anthropogenic-induced global climate change reduction has, therefore, forceful links with GHG emissions accounting and reporting. The latter is a broad set of disclosures dealing with the impact of human and corporate activities on the climate, e.g., carbon emissions disclosures, other greenhouse gas (GHG) emissions disclosures, and footprint-related disclosures. The environmental issue is relevant not only for governments, but also for public and private organizations from different activity sectors [3]. Companies (especially large ones) are responsible for releasing great quantities of polluting emissions into the atmosphere. It is reasonable to expect them to be widely involved in the reduction of such emissions, in the reporting of the activities carried out to achieve this goal, and in the results actually achieved. Consistent with these expectations, the papers analyzed in this literature mainly focused on accounting and reporting for $CO_2$ and GHG emissions.

While corporations and the accounting profession are moving towards the need to provide a climate change disclosure, which can assist public decision-making [2], academic research on this topic is still underdeveloped [4,5]. As a consequence, there are both challenges and an opportunities to undertake urgently needed research into a broad range of accounting and accountability issues about climate change, GHG emissions accounting, reporting, assurance, emissions management, and GHG reductions [6].

The authors of this paper performed a systematic literature review on the climate change accounting-based literature during the period 1999–2018, by focusing on articles published in international peer reviewed journals. The objective was to map the conceptual structure and the evolution of the climate change-related accounting and reporting literature in order to:

- Understand its current status;
- Identify gaps to be filled;
- Challenge scholars to increase their work on relevant areas for their future sustainability accounting research.

This paper can be useful for a wide array of readers because it:

- Shows the different perspectives addressed by climate change accounting and reporting research;
- Points out new research areas in which sustainability accounting and reporting should be beneficial;
- Identifies some perspectives of development of the accounting discipline.
- In fact, "The margins of accounting change as the boundaries of accounting are redrawn. The margins are fluid and mobile, rather than static. What is on the margins at one point in time can become central or taken-for-granted, relatively fixed and durable, at a later date. Moreover, the margins of accounting vary from one national setting to another" [7]. In this context, accounting and reporting for SDGs, and more in detail to climate change-related purposes, have only become the focus of research in the accounting discipline recently.

Until a few decades ago, accounting was typically understood as financial accounting or management accounting, focusing essentially on documenting the economic and financial performances of companies. The evolution of the information needs of corporate stakeholders is progressively expanding the quantities and types of information required for the disclosure of companies.

This paper is structured as follows: Section 2 shows the research background, Section 3 describes the research method, Section 4 focuses on the findings, Section 5 discusses the findings, Section 6 presents the main conclusions, and Section 7 discusses future trajectories of climate change-related accounting literature and suggests future developments for climate change-related accounting purposes.

## 2. Research Background

The study of accounting is becoming more multidisciplinary [8]. Scholars analyzing the relationships between accounting and society argue that accounting is important not only for a single firm but also in more general social processes [9]. What is of interest in this paper are the close connections between accounting and social processes [10]. In the context of climate change, it is interesting to analyze how accounting scholars respond to growing societal concerns about climate change, but it is also interesting to examine the role accountants could have in influencing the common perception of this problem and how to deal with it.

As Hopwood and Miller [11] suggested, "Accounting could not and should not be studied as an organizational practice in isolation from the wider social and institutional context in which it operates." This means that social processes shape and are shaped by accounting.

The need to consider the natural environment in accounting decisions was felt starting in the 1960s and 1970s [12], giving rise to environmental accounting [13]. Starting in the late 1980s and early 1990s, Reference [14] suggested that the accounting literature and practice should include environmental and social aspects, such as waste and energy reporting, compliance and ethical audits, social and environmental reporting, environmental impact assessment, and accounting for environmental assets and liabilities. In the 1990s, Elkington [15] introduced the concept of 'triple bottom-line' (TBL), arguing the need for companies to report not only on financial, but also on their social and environmental performance.

In 1997, the United Nations Environment Program (UNEP) and the US-based nongovernmental organization Coalition for Environmentally Responsible Economies (CERES) launched the Global Reporting Initiative (GRI), aimed at improving the quality and utility of TBL. The GRI provided certified software and tools to assist with data collection and report preparation, standard templates, and checklists.

Subsequently, owing to carbon and greenhouse emission legislation, there emerged internal carbon and greenhouse gas accounting, used to determine the liabilities of companies to the accounting of tradable rights arising from emissions taxes and emissions trading schemes [16,17], and to report on GHG emissions.

In recent decades, public and private organizations have been expected to be more and more accountable for the impact of their actions on the environment. Stakeholders expect, in particular, information on the impact of organizations' actions on climate change [3,18]. In this context, there was

an evolution of sustainability and climate change-related information provided by public and private organizations, with it increasing especially since the issue of the Kyoto protocol in 1997.

In the 1980s and 1990s, environmental information was mainly provided in the annual reports of companies operating in different countries [3]. Thereafter, environmental information was usually provided through voluntary information displayed in sustainability reports published on the organizations' websites. This information often incorporated GRI indicators, as well as other economic, environmental, and social features [3]. As a consequence, sustainability reporting has gained relevance as a means to be accountable for the environmental impacts of organizations' activity. The need for high-quality reporting about sustainable development, SDGs, and financial balance was highlighted during the 32nd session of the Intergovernmental Working Group of Experts in International Standards of Accounting and Reporting, held in Geneva in 2015 [19]. Accounting practitioners and scholars play a relevant role in the achievement of the SDGs (and of climate change-related objectives), since they are involved in accounting, reporting, and auditing activities related to these goals.

The disclosure of climate change-related information falls within a "grey area" between carbon/GHG financial accounting and nonfinancial disclosure (the latter is also known as "narrative disclosure") [9]. In this context, academic research about climate change is focusing both on sustainability reporting, which has now become a recognized research area in accounting studies [5], and on the integration of climate change reporting into financial reports [9,20].

According to Haque and Islam [21], the pressures behind climate change accountability and disclosure come from various stakeholders, i.e., government bodies (regulators), institutional investors, environmental NGOs, and media accounting professionals.

Reporting to government bodies (i.e., regulators) is mainly aimed at fulfilling obligations connected to the legislation on emissions rights (tradable emission rights and obligations related to emission trading schemes). Reporting in terms of corporate responsibilities and commitments is mainly aimed at supporting the image and reputation of the organization towards various types of stakeholders.

According to Solomon et al. [5], among the stakeholder pressures, climate change reporting is mainly driven by risk and risk management motivation. In other words, the need to disclose climate change information is induced by institutional investors' (who are the main financial stakeholders of corporations) belief that climate change: (a) is a relevant risk; (b) is the most important issue within sustainability; (c) is a relevant aspect for organizations' clients, who need climate change-related risks to be managed in their portfolio investment.

To date, the studies on accounting and society have not been extensively applied to the focus of sustainable development. Some authors started dealing with the relationship between accounting and SDGs [22–24], focusing on:

- SDG no. 6 (clean water and sanitation), e.g., [25,26]. Ref. [25] focused on the intersection between accounting and human rights. Within the latter, the author questioned if the access to information can be considered as a human right, focusing specifically on environmental information related to water sources. Ref. [26] focused on the reporting of sustainability information to public water companies' stakeholders.
- SDGs no. 5 (gender equality), SDG no. 10 (reduced inequalities), and 16 (peace, justice and strong institutions), e.g., [27–30]. Among this literature, it is worth mentioning [28], containing an introductory analysis of the potential role of the International Accounting Standards Board (IASB) for the application of relevant human rights norms, and [30], dealing with social accounting research related to economic inequality, and calling for a deep involvement of social accountants in public debates about the future of resource distribution.
- SDGs no. 14 and no. 15 (life below the water/life on land), e.g., [31–35]. Among this literature, it is worth mentioning [32], dealing with the potential role of environmental accounting in the reconstruction of the knowledge of the social, economic, and environmental risks of salmon farming and in the choice of adopting organic production methods. Ref. [34] extended the research

stream of biodiversity accounting, trying to verify if Jones' natural inventory model is applicable to a specific local context.

Climate change studies on SDG 13 have focused on carbon and greenhouse gases accounting/reporting. Stechemesser and Guenther [36] proposed a definition of carbon accounting obtained through a systematic literature review, including many perspectives and research streams. Bebbington and Larrinaga-Gonzalez [37] focused on accounting issues, with specific regard to the problems related to the valuation of pollution allowances, their identification as assets, and the liabilities connected with pollution. The paper also focused on the risks and uncertainties arising from global climate change. Ascui and Lovell [38] reviewed the literature on carbon accounting and provided a definition including the different meanings encompassed by the term. Kolk, Levy, and Pinkse [39] analyzed the corporate responses to climate change through the development of reporting mechanisms for greenhouse gases, focusing on carbon disclosure. Brander [40] described the differences between 'attributional' and 'consequential' greenhouse gas accounting methods. Comyns and Figge [41] investigated 45 greenhouse gas (GHG) reports of oil and gas companies 1998–2010. Findings showed that quality score, obtained considering the dimensions of accuracy, completeness, consistency, credibility, relevance, timeliness and transparency, had not significantly improved in the analyzed period. Milne and Grubnic [6] focused on carbon emissions and GHG footprints behavior. Results showed that accounting for carbon and GHG emissions is challenging due to uncertainties in estimation techniques.

Pellegrino and Lodhia [42] focused on climate change accounting and carbon accounting, i.e., a subset of sustainability accounting. Ramírez and González [43] investigated how the financial statements are affected by the obligations of companies to control and compensate their carbon emissions. Linnenluecke, Birt, and Griffiths [24] suggested that accounting can support organizational climate change adaptation through: (i) the assessment of vulnerability and adaptive capacity, (ii) the valuation of adaptation costs and benefits, and (iii) the disclosure of the risks associated with climate change impacts. Le Breton and Aggeri [44] showed how the development and dissemination of carbon accounting tools in a public organization can affect the actions of companies. Herbohn et al. [45] and Clarkson et al. [46] observed high-polluting companies starting to address carbon reduction, energy efficiencies, and greenhouse gas emission reduction efforts.

The authors of this paper focused specifically on the existing accounting research about the SDG13. They undertook a systematic literature review, with the twofold aim to highlight the current state of the existing research and the future research directions on this issue.

To the best of our knowledge, there is only one previous study on the same theme, [21], that reviews the contributions of a special issue of Sustainability Accounting, Management, and Policy Journal (no. 3): Carbon accounting, management, and policy. In this paper, four avenues for research were proposed: climate change as a systemic and social issue, the multi-layered transition apparatus for climate change, climate vulnerability, and the future of carbon accounting. Another literature review focused on the definition of carbon accounting [36]. Based on the results of their review, the authors proposed a definition that can be used not only for academic purposes, but also by practitioners and by legislators.

The focus of the review here presented is broader than the above-mentioned reviews because it focuses not only on accounting and climate change, but also on reporting. The economic and competitive consequences of sustainability issues cannot be overlooked [26,47]. This implies that sustainability issues obtain sufficient consideration in information management accounting and reporting. To this end, conventional corporate accounting systems should incorporate environmental, social, and financial issues. In this context, new development paths for accounting research and practice unfold, with particular reference to accountability methods and tools. GHG concentrations and climate change have ecological, social, and economic effects that must be taken into account in corporate communication [27].

The emergence of climate change as a technical issue for accounting scholars and practitioners can be dated back from around 2000. At the time, accountants deemed their role should be essentially technical and nonstrategic [47].

Starting in 2005, significant changes have occurred in accountants' approach to climate change problems. In this period, several initiatives related to climate change accounting/accountability emerged, e.g., an increase in the numbers of climate change reports, newsletters, and other initiatives by accounting professional bodies. In 2008, the International Accounting Standards Board (IASB) intervened on this issue in collaboration with the US Financial Accounting Standards Board (FASB).

Climate accounting [48], climate change accounting [6,9,49,50], accounting for climate change [2,6], and climatic accounting are expressions used in the literature. However, there is no categorical definition of them [51]. Hirschfeld et al. [48] defined climate accounting as "Accounting for the impact on the climate of [ … ] production processes" aimed at comparing "the effects on the climate of different [ … ] production processes" (p. 11). According to Schaltegger and Csutora [52], while carbon accounting deals only with carbon emissions, climate change accounting deals also with further GHG emissions. According to Ngwakwe [49], climate change accounting relates to emission accounting, GHG footprinting, carbon capture and storage, and sequestration calculations. According to Brown et al. [53], accounting of climate change deals with "reviews of climate change reporting, stakeholder reactions to disclosures of climate change information, new systems of accounting designed to incorporate climate change performance, discussion about the role of accounting in promoting or undermining the climate change, environmental audits, discussion about general climate change conditions, climate change accounting policies, climate change coverage of products and processes-related information, climate change financial related data, sustainability, environmental aesthetics, development of theories to explain or inform climate change accounting practices, and discussion of methods and methodological issues associated with this research". Evangelinos et al. [54] distinguished among formal/informal accounting methods and financial/nonfinancial methods.

According to the measurement units, informal accounting methods can be divided into methods using financial terms (e.g., carbon cost accounting and full cost accounting methods) [55]; methods using nonfinancial terms, including life cycle assessment and eco-balance [56,57]; reporting methods (e.g., the Carbon Disclosure Project) [20]; and eco-efficiency accounting methods. There is a broad agreement on the importance of environmental information (in financial and nonfinancial terms) for the measurement of corporate GHG and economic performance [47,58].

Even if there is a general agreement about the importance of these measurements for the evaluation of the performance of corporations, there is a lack of frameworks based on accounting principles to frame them.

Evangelinos et al. [54] concluded their study by suggesting that currently, a large part of accounting approaches accounts for information on climate change on a voluntary basis. The existence of informal accounting standards leads to untrustworthiness and awkwardness in climate change accounting and accountability.

According to Ilinitch et al. [59] and Nikolaou and Evangelinos [10], the lack of generally accepted guidelines for environmental information accounting and accountability is a significant problem. Hopwood [60] identified analogous difficulties in carbon accounting. As a result, there is ineffectiveness in stakeholders' decision processes, in the measurement of corporate environmental performance, and in public policy entities [61].

## 3. Materials and Methods

The authors of this paper used a systematic literature review approach [62]. This approach was chosen since: (a) it allows the performance of a located and synthesized research on a specific question, using structured, clear, and replicable procedures [24] and (b) compared to conventional narrative reviews, it has more rigor in the research process [63].

According to Fink [64], Tranfield et al. [65], and Stechemesser and Guenther [36], the review was structured into four main phases:

(1) Selection of research questions, bibliographic database, and search terms. During this phase, the authors also selected a tool to perform science mapping analysis;

(2)    Definition and use of review criteria for the inclusion/exclusion of the relevant literature;

(3)    Development and application of a methodological review protocol;

(4)    Synthesis of the findings.

The first three phases are described in the subsections below, while the fourth phase is described in Sections 4 and 5.

### 3.1. Selection of Research Questions, Science Mapping Analysis Tool, Bibliographic Database, and Search Terms

During phase 1, the authors identified the following general research questions (indicated below as RQs):

(a)    What is the knowledge base of climate change accounting/reporting and what is its intellectual structure (RQ1)?

(b)    What is the research front (or conceptual structure) of this research field (RQ2)?

(c)    What is the social network structure of the scientific community writing about this topic (RQ3)?

(d)    What lessons can be learned from the literature analyzed about the current state and the future developments of the climate change accounting and reporting-related literature (RQ4)?

According to Kitchenham and Charters [66], the selection of the RQs is the most important part of any systematic review. The first three RQs above mentioned are general types of questions that can be answered using science mapping [67], while RQ4 was addressed through qualitative analysis.

During this phase, the authors also identified a tool to perform science mapping analysis: bibliometrix [67]. This is an open source tool allowing bibliometric analyses. Bibliometrics can be understood as "the application of mathematical and statistical methods to books or other media of communication" [68]. It helps researchers to apply an organized, clear, and reproducible review process based on a statistical weighing of science, scientists or scientific action [67,69,70]. Bibliometrix can process data retrieved from WoS or Scopus. According to the creators of bibliometrix, WoS is the most suited database to retrieve data in the accounting field. Therefore, the authors of this paper chose this bibliographic database instead of other similar databases (e.g., Scopus, Science Direct, or Google Scholar).

During phase 1, the authors also chose the search terms. At a first stage (here identified as subphase 1A), the objective was to identify and analyze what accountants (both on the academic and the professional level) publish about sustainability. Within this subphase, the authors therefore chose a complex set of keywords through the following topical query:

-    (("account *" OR "report *") AND ("global reporting initiative" OR "GRI"));

-    (("account *" OR "report *") AND ("social *" OR "environment *" OR "sustainab *" OR "CSR" OR "responsib *" OR "TBL" OR "triple" OR "integr *"));

-    (("account *" OR "report *") AND (* financial));

-    (("account *" OR "report *") AND ("carbon" OR "divestment" OR "Paris agreement" OR "multigenerational benefit" OR "climate change" OR "consequential" OR "natural disaster" OR "ecological").

This keywords search showed more subfields in which accounting/reporting and sustainability were the object of the scientific studies. Among the many topics, in the following research step (here identified as subphase 1B), the researchers focused on accounting/reporting and climate change, which is an emerging field of interest for accounting professionals [2] and academics.

### 3.2. Definition and Use of Review Criteria for the Inclusion/Exclusion of the Relevant Literature

The first results that emerged from the keywords search launched in the subphase 1A were refined using the following criteria:

- Document type: only articles (with the exclusion of presentations, book reviews, comments, and patents);
- Languages: only English, in order to avoid a language bias [36];
- Research areas: environmental sciences, ecology, business economics, operations research management science, energy fuels, social science's other topics, water resources;
- Web of Science categories: environmental sciences, ecology, energy fuels, engineering environmental, environmental studies, management, water resources, business finance, business, social sciences interdisciplinary, green sustainable science technology, operations research management science, development studies, and ethics;
- Timespan: 1999–2018. The authors used 1999, as the starting year of the analysis since in this period the first sustainability reports and similar reporting tools were published as a result of the issuing of the first GRI guidelines [7];
- Citation Indexes: SCI-EXPANDED, SSCI, A&HCI, CPCI-S, CPCI-SSH, ESCI.

These search criteria did not restrict the search to a few journals (typically about accounting topics), but broadened the field of investigation to a wide range of scientific journals operating in various sectors. As a result of this broader search, approximately 99,000 relevant documents were found. Two of the researchers performed an individual analysis of each of the documents to decide to include them within the analysis (if judged to be coherent with the research project, i.e., if explicitly focusing on accounting/reporting and sustainability) or to exclude them (if they were judged not to be coherent with the research project). In the event of disagreement between the two researchers, they discussed each doubtful case, and decided by common agreement whether to include or exclude them. In this way, the two researchers selected a sample of approximately 900 articles about "accounting/reporting and sustainability".

Thereafter, the researchers reassessed the approximately 900 articles above mentioned, in order to focus only on articles dealing with climate change accounting/reporting. To this end, they searched the keywords "climate", "climate change", "climate-change", and "warming" in the following fields: (a) title, (b) authors' keywords, (c) keywords plus (keywords plus are keywords assigned to a paper by a Clarivate Analytics' algorithm. They are words or phrases frequently appearing in the titles of the references of an article, but which do not appear in the title of the article itself. They are considered to be more significant than authors' keywords in order to identify the contents of the article. In fact, the authors' keywords are not always clear enough to explain the main content of an article), and (d) abstract. In this way, they selected 148 documents.

The use of these review criteria produced the risk of underestimating the first sample extracted during subphase 1A, as it was possible that the former sample contained articles dealing with issues related to accounting for climate change (e.g., carbon accounting/reporting/disclosure/assurance, GHG accounting/reporting/disclosure/assurance, footprint disclosures, etc.) but did not explicitly use "climate change" in the title, in the keywords, or in the abstract. In fact, accounting for climate change falls within a wide range of topics dealing with social and environmental issues. Considering the wide array of keywords used in subphase 1A, the authors decided that the risk of underestimating the second sample can be considered to be acceptable, because this search mode allowed the researchers to focus exclusively on publications explicitly mentioning aspects related to climate change.

At this stage, three researchers analyzed the selected papers by dividing them into coherent with subphase 1B (to be included) or not coherent (to be excluded). In the absence of perfect agreement, the researchers decided by majority vote whether to include or to exclude individual publications. In this way 85 papers were selected that addressed "accounting/reporting and climate change". Therefore, this paper is focused upon the articles found during subphase 1B, which are listed in Appendix A.

### 3.3. Development and Application of a Methodological Review Protocol

In any systematic review, the development and application of a review protocol is an important step [66]. The review protocol used in this paper is described in Table 1. At the end of the bibliometric analyses, the authors adopted a qualitative coding scheme in order to analyze the contents of the articles within the sample. During a pilot stage, two authors read five of the papers included in the sample (using both abstracts and full texts) and thereafter discussed them and made preliminary classifications. They adopted the Guthrie et al. [71] and the Dumay et al. [72] coding schemes, which were consequently slightly modified, as outlined in Table 2. Thereafter, one author coded the papers manually and the other two authors checked the coding. In the case of disagreement among the authors, the coding result was decided by majority vote.

**Table 1.** The review protocol.

| *Sections* | *Questions* | *Answers: Open/Closed (Codified)* |
|---|---|---|
| **Bibliographic data** | | |
| Author(s) | Who is/are the author(s) of the publication? | Open |
| Year | In which year was the work published? | Open |
| Title | What is the title of the publication? | Open |
| Authors' affiliation | What is the university affiliation of the author(s)? | Open |
| Type of publication | What kind of publication? | Open |
| Journal name | If it is a journal: what is the journal's name? | Open |
| **Research framework** | | |
| A. Jurisdiction | What is the context of the publication? | A1. Supranational/international comparative<br>A2. National<br>A3. Local government<br>A4. Public Business Enterprise<br>A5. Private company<br>A6. Specific economic sector<br>A7. One organization<br>A8. Other |
| B. Location | Which is the specific location of the publication? | B1. Europe<br>B2. North America<br>B3. South America<br>B4. Australasia<br>B5. Asia/China<br>B6. Africa<br>B7. Other (e.g., more locations, nowhere in particular) |
| C. Focus of the article | On which research field the article is focused? | C1. External reporting<br>C2. Auditing<br>C3. Accounting<br>C4. Governance<br>C5. Management control/strategy<br>C6. Performance measurement<br>C7. Budgeting<br>C8. Other |
| D. Research methods | Which research method is used in the publication? | D1. Case/field study/interviews<br>D2. Content analysis/historical analysis<br>D3. Survey/questionnaire/other empirical<br>D4. Commentary/normative/policy<br>D5. Theoretical literature review/empirical |
| E. Academic or practitioner author(s) | Which is the institutional background of the author(s)? | E1. Only academic(s)<br>E2. Only practitioner(s)<br>E3. Academic(s) and practitioner(s) |

The review protocol developed by the authors included two parts:

(1) In the first part there were the bibliographic data of each article falling within the final sample (i.e., author(s)' names and affiliations, year and title of publication, journal's name, keywords and abstracts);

(2) In the second part there was the coding scheme, divided into the following criteria: (A) jurisdiction, (B) location, (C) focus of the article, (D) research methods, (E) academic or practitioner author(s).

The fourth step of the systematic review "synthesis of the findings" is discussed in Section 4 (results) and Section 5 (discussion).

## 4. Results

Bibliometrix was used to perform descriptive analyses of the bibliographic data. These analyses used the following functions [67]:

(1) the "biblioAlanalysis" function, which calculated the bibliometric measures displayed in Section 4.1;

(2) the functions "summary"and "plot", which summarized the main results that emerged from the bibliometric analysis (described in Section 4.2);

(3) the "citations" function (described in Section 4.3).

In addition to the first three steps, classified as "descriptive analyses", there was the last step classified as the "network analysis" (described in Section 4.4).

Additionally, a qualitative analysis was performed outside the bibliometrix functions, including: (a) the Guthrie et al. [71] and the Dumay et al. [72] analytical frameworks and (b) the examination of the scientific background of the most relevant authors of the sample. These results are described in Section 4.5.

### 4.1. Main Information Derived from the Collection of Selected Documents

The literature review was performed on 85 papers published in the period 1999–2018, composed of 83 articles and two conference proceedings papers published in 28 different journals. The articles were written by 233 authors, of which only nine papers were single-authored. Table 2 summarizes the main information about the collection.

**Table 2.** Main information about the 85 articles selected for this research.

| % | |
| --- | --- |
| **Description** | **Results** |
| Documents | 85 |
| Sources (Journals, Books, etc.) | 28 |
| Keywords Plus (ID) | 268 |
| Author's Keywords (DE) | 275 |
| Period | 1999–2018 |
| Average citations per documents | 30.48 |
| Authors | 233 |
| Author Appearances | 249 |
| Authors of single-authored documents | 9 |
| Authors of multi-authored documents | 224 |
| Single-authored documents | 10 |
| Documents per Author | 0.36 |
| Authors per Document | 2.74 |
| Co-Authors per Documents | 2.93 |
| Collaboration Index | 2.99 |
| | |
| Document types | |
| ARTICLE | 83 |
| ARTICLE; PROCEEDINGS PAPER | 2 |

### 4.2. Main Results Derived from the Bibliometric Analyses

The functions summary and plot summarize the main results that emerged from the bibliometric analyses. In this section, the results presented relate to: annual scientific production, most relevant and cited sources, most relevant authors, most productive countries based on first author's affiliations, and most frequent keywords.

Looking at the annual scientific production during the period 1999–2017, although irregular, there was a clear, increasing trend in the number of publications per year. This trend was most evident starting in year 2007. Coincidently, between 2006 and 2009, environmental issues were the subject of a strong international debate. This period of deep attention towards environmental issues was followed by events that drew attention to environmental issues, such as the economic recession, bank failures, and potential EU member state failures [6]. The most productive year was 2018 (18 articles), whereas there was a significant decrease in publications in 2017 (only three articles).

Figure 1 shows the 20 "most relevant sources" (using bibliometrix terminology), i.e., the journals in which most of the articles analyzed were published. The journals in which the greatest number of scientific articles were published was the Journal of Cleaner Production and Accounting Auditing and Accountability Journal. It is interesting to note that among the twenty "most relevant sources" displayed in Figure 1, only a few of them are accounting journals. In most cases, the journals publishing the highest number of papers within the sample are specialized in sustainability issues. This shows that sustainability journals publish contributions from different research areas, including accounting, while accounting journals seem more reluctant to accept contributions on sustainability issues, perhaps since they are more focused on "traditional" themes within the accounting discipline.

**Figure 1.** Most relevant sources.

Note that not all the "most relevant sources" (Figure 1) coincide with the most cited sources (Figure 2). In fact, while the Journal of Cleaner Production was the source of the most articles, the articles that were the most frequently cited were published in Accounting, Organizations, and Society, which is absent from the ranking of "most relevant sources".

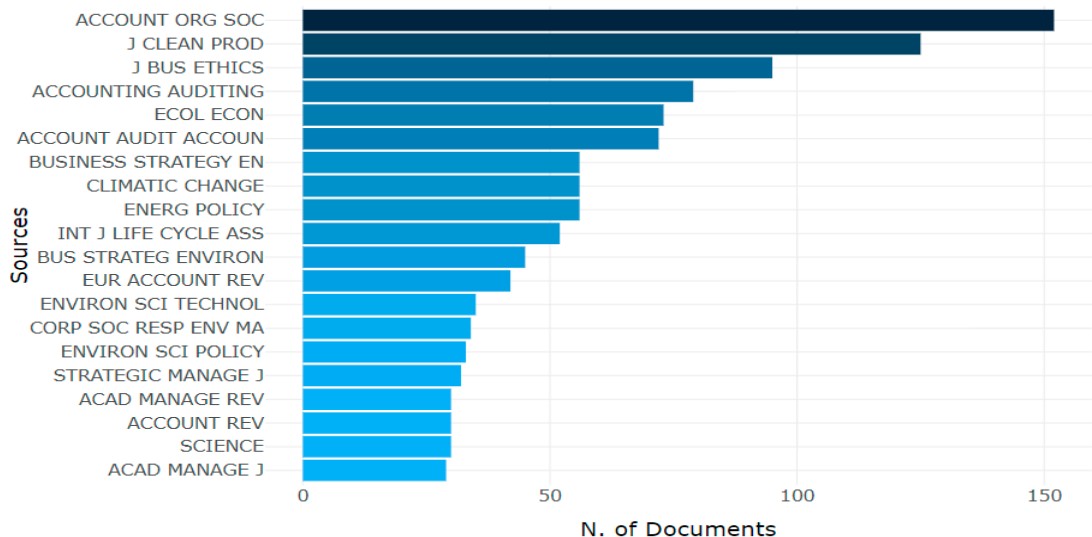

**Figure 2.** Most cited sources.

Table 3 shows the "most relevant authors" (using bibliometrix terminology), i.e., the most active authors in terms of articles and of fractionalized articles (the latter is obtained by setting the number of publications per author by the number of co-authors per publication). In terms of publications number, the most active author was Stefan Schaltegger, affiliated with the Centre for Sustainability Management (CSM) at Leuphana University (Germany), who published three articles within the sample. His most recent article was published in 2015. Considering the contribution of each author to the preparation of the article (articles fractionalized), the ranking was led by Prof. Matthew Brander—Senior Lecturer in Carbon Accounting and Director, University of Edinburgh—who published two single-authored documents between 2016 and 2017.

**Table 3.** Most relevant authors.

| Authors | Articles | Authors—Fractionalized | Articles—Fractionalized |
|---|---|---|---|
| Schaltegger S. | 3 | Brander M. | 2.0 |
| Ascui F. | 2 | Mozner Z.V. | 1.5 |
| Ben amar W. | 2 | Schaltegger S. | 1.1 |
| Brander M. | 2 | Alvarez I.G. | 1.0 |
| Burritt R.L. | 2 | Ascui F. | 1.0 |
| Cowie A.L. | 2 | Birdsey R.A. | 1.0 |
| Csutora M. | 2 | Comyns B. | 1.0 |
| Gallego alvarez I. | 2 | Csutora M. | 1.0 |
| Garcia sanchez I.M. | 2 | Gillenwater M. | 1.0 |
| Gillett N.P. | 2 | Haripriya G.S. | 1.0 |
| Kirschbaum M.U.F. | 2 | Hasselmann K. | 1.0 |
| Lodhia S. | 2 | Lee K.H. | 1.0 |
| Lovell H. | 2 | Lodhia S. | 1.0 |
| Mcilkenny P. | 2 | Lovell H. | 1.0 |
| Mozner Z.V. | 2 | Ben amar W. | 0.8 |

In the same sample, 14 authors published two articles about climate change accounting; the remaining authors published only one article about this topic. The low number of publications on climate change accounting per author deserves reflection on the reasons of this situation. It is reasonable to assume that this is still an underdeveloped research field. By analyzing scientific backgrounds and affiliations of the most 15 productive authors (this analysis was conducted outside the bibliometrix package and is described in Section 4.5), the authors of this paper found that almost half of them were accounting academics who were regularly dealing with climate change accounting-related topics.

Many other authors of articles in the selected group were non-accounting academics (e.g., scholars of physics, biology, economics) regularly dealing with climate change-related topics. Focusing on the authors' affiliations, the primacy of the University of Edinburgh emerged, while the University of South Australia was in second place.

Table 4 shows the most productive countries based on first author's affiliation. The largest number of articles from among the 85 of this research were published by authors in the UK (28 articles), followed by Australia (27), USA (18), Germany (15), and Canada (14). Table 4 can be compared to the collaboration networks among countries (whose figure has not been presented here). There are two main collaboration networks including some of the "most productive countries" of Table 4. In the first one there are authors from the UK, the USA, Canada, and Sweden. In the second one, there are authors from Australia, New Zealand, and the Netherlands.

**Table 4.** Most productive countries.

| Region | Freq | Region | Freq |
|--------|------|--------|------|
| UK | 28 | Finland | 5 |
| Australia | 27 | Greece | 5 |
| USA | 18 | Netherlands | 5 |
| Germany | 15 | Austria | 4 |
| Canada | 14 | Denmark | 4 |
| France | 12 | Hungary | 4 |
| China | 8 | New Zealand | 4 |
| Brazil | 7 | Norway | 4 |
| Spain | 7 | Italy | 3 |
| Sweden | 7 | Argentina | 2 |

Table 5 shows the most frequently used keywords, in terms of keywords plus (identified by Clarivate Analytics), authors' keywords, words in the titles, and words in the abstracts.

As already explained, keywords plus are words or phrases frequently appearing in the titles of the references of an article, but which do not appear in the title of the article itself. In this, they differ from authors keywords, words in titles, and words in abstracts, in that the latter three are words chosen by the authors of the papers. The creators of bibliometrix consider keywords plus as the most significant keywords in bibliometric analysis, as the keywords chosen by the authors are not always clear enough to explain the main content of an article. Anyway, the words selected by authors also deserve reflection.

As could have been expected from an analysis focused on climate change, the words "climate", "change", and "climate change" are among those most frequently occurring in all four types of words investigated in Table 5. The concepts of "emissions" and "disclosure/reporting" also occur frequently in all the dimensions analyzed in Table 5.

Within the analyzed papers, the concepts of "accounting" and "reporting" occur more frequently in the titles of publications and in their abstracts than in the keywords plus and in the authors' keywords. This situation can be interpreted on the basis of the contents of the articles analyzed. In many cases, it was not simple to categorize some articles of the sample as falling within the accounting and reporting issues or on the margins of/outside them. This happened, for instance, in some papers dealing with measurement/quantification of pollutant emissions which, however, cannot always easily be traced back to the accounting discipline. In various cases of this kind, the authors wondered if it was more correct to classify some of the papers analyzed in a context of "to count" (i.e., counting the amount of emissions in the environment) instead of "to account".

**Table 5.** Most frequently used keywords.

| Keywords Plus (ID) | Occ. | Authors Keywords (DE) | Occ. | Titles | Occ. | Abstracts | Occ. |
|---|---|---|---|---|---|---|---|
| Climate change | 16 | Climate change | 27 | Accounting | 47 | Carbon | 229 |
| Emissions | 15 | Carbon accounting | 14 | Carbon | 41 | Climate | 196 |
| Companies | 11 | Greenhouse gas emissions | 6 | Climate | 22 | Accounting | 173 |
| Environmental disclosures | 8 | Ghg emissions | 5 | Greenhouse | 16 | Emissions | 172 |
| Greenhouse gas emissions | 8 | Sustainability | 5 | Change | 15 | Change | 159 |
| Performance | 8 | Sustainable development | 5 | Gas | 15 | Reporting | 99 |
| Legitimacy | 7 | Disclosure | 4 | Reporting | 15 | Ghg | 86 |
| Strategies | 7 | Global warming | 4 | Corporate | 13 | Paper | 72 |
| Co2 | 6 | Greenhouse gas | 4 | Emissions | 13 | Disclosure | 70 |
| Management | 6 | Kyoto protocol | 4 | Disclosure | 12 | Environmental | 70 |
| Sustainability | 6 | Sustainability reporting | 4 | Assessment | 7 | Companies | 67 |
| Determinants | 5 | Carbon | 3 | Companies | 7 | Greenhouse | 57 |
| Impact | 5 | Carbon cycle | 3 | Emissions | 7 | Firms | 54 |
| Sequestration | 5 | Carbon disclosure | 3 | Ghg | 7 | Gas | 52 |
| Social responsibility | 5 | Carbon stocks | 3 | Management | 7 | Management | 50 |
| Corporate social responsibility | 4 | Climate change mitigation | 3 | Performance | 7 | Global | 49 |
| Energy | 4 | Emissions | 3 | Sustainability | 7 | Analysis | 46 |
| Financial performance | 4 | Environmental accounting | 3 | Case | 6 | Corporate | 43 |
| Governance | 4 | Environmental reporting | 3 | Energy | 6 | Study | 43 |
| International trade | 4 | G.h.g. accounting | 3 | Environmental | 6 | Methods | 41 |

The literature review revealed a typical aspect of scientific disciplines, that is, their gradual evolution over time. The margins of scientific disciplines are not fixed, but rather flexible, and change according to geographical, cultural, and temporal variables [73]. This is true also for accounting, whose area of investigation has been expanding in recent decades. The need to report corporate performance to an ever-growing audience of stakeholders implies the expansion of the aspects under which these performances are analyzed. In addition to financial performance, environmental and social performance need to be reported. In this logic, the accounting margins widen to meet new knowledge needs of the corporate stakeholders. For this reason, the concept of "reporting" no longer includes only the financial statements, but also various nonfinancial reporting tools. For the same reason, the concept of "accounting" is expanding, including aspects other than traditional financial, cost, and management accounting. The literature review was therefore useful to understand "what counts as accounting" in the context of climate change and to highlight the expansion of the accounting discipline towards "nontraditional" forms of accounting and reporting.

*4.3. Information about Citations*

Bibliometrix provides some information about the citations. Table 6 shows the 20 most-cited papers in terms of total citations and total citations per year. The five most-cited papers are:

(1) Kolk, Levy and Pinske, 2008 [Appendix A], dealing with the institutionalization of carbon disclosure as a corporate response to emerging climate change information needs.
(2) Lozano and Huisingh, 2001 [Appendix A], focusing on sustainability reporting and on the guidelines and standards addressing sustainability issues;
(3) Barrett, 2013 [Appendix A], containing a case study about consumption-based GHG emission accounting in UK;
(4) Bebbington and Larrinaga-Gonzalez, 2008 [Appendix A], focusing on the problems connected with the valuation of pollution allowances and their identification as assets and liabilities;
(5) Brandao, Levasseur, Kirschbaum, Weidema, Jorgensen, Hauschild, Pennington, and Chomkhamsri, 2013 [Appendix A], dealing with life cycle assessment (LCA) and carbon footprinting (CF) as tools for the environmental assessment of products. It analyzes some methods for accounting the potential climate impacts of carbon sequestration.

The most-cited papers were written by accounting and nonaccounting authors and were published in both accounting and non-accounting journals.

**Table 6.** Top 20 most-cited papers.

| Paper | Total Cit. (TC) | TC per Year |
|---|---|---|
| Kolk and Pinske, 2008, European Accounting Review | 241 | 21.9 |
| Lozano and Huisingh, 2011, Journal of Cleaner Production | 196 | 24.5 |
| Barrett, 2013, Climate Policy | 123 | 20.5 |
| Bebbington and Larrinaga-Gonzalez, 2008, European Accounting Review | 110 | 10.0 |
| Brandao et al., 2013, International Journal of Life Cycle Assessment | 108 | 18.0 |
| Rankin, Windsor and Wahyuni, 2011, Accounting, Auditing, and Accountability journal | 86 | 10.7 |
| Schaltegger and Csutora, 2012 Journal of Cleaner Production | 84 | 12.0 |
| Williams and Schaefer, 2013, Business Strategy and the Environment | 78 | 13.0 |
| Prado-Lorenzo, Rodriguez Dominguez, Gallego Alvarez and Garcia Sanchez, 2009, Management Decision | 75 | 7.5 |
| Burritt, Schaltegger and Zvezdov, 2011, Australian Accounting Review | 67 | 8.4 |
| Gentil, Christensen and Aoustin, 2009, Waste Management and Research | 64 | 6.4 |
| Pellegrino and Lodhia, 2012 Journal of Cleaner Production | 58 | 8.3 |
| Ascui and Lovell, 2011, Accounting, Auditing, and Accountability Journal | 53 | 6.6 |
| Haddock Fraser and Tourelle, 2010, Business Strategy and the Environment | 49 | 5.4 |
| Solomon, Solomon, Norton and Joseph, 2011, Accounting, Auditing, and Accountability Journal | 48 | 6.0 |
| Lee, 2012, Journal of Cleaner Production | 48 | 6.8 |
| Gillett and Matthews, 2010, Environmental Research Letters | 44 | 4.9 |
| Milne and Grubnic, 2011, Accounting, Auditing, and Accountability Journal | 42 | 5.2 |
| Whittaker, McManus and Smith, 2013, Environmental Modelling and Software | 40 | 6.7 |
| Sullivan and Gouldson, 2012, Journal of Cleaner Production | 39 | 5.6 |

*4.4. Network Analysis*

Bibliometrix helped the authors to conduct some types of "network analyses", i.e., analyses involving connections among different parameters under scrutiny. In this paper, the following types of "network analyses" [67] were performed:

(a) Co-citation: this is the case in which two articles are both cited in a third article;

(b) Co-words: this type of analysis helps researchers to depict the conceptual structure of a framework through a word co-occurrence network in order to map and classify words extracted from titles, keywords, or abstracts of a given bibliographic collection;

(c) Collaboration: i.e., networks whose nodes are the authors and whose links are the co-authorships.

A selection of the main network results is presented in this section. First of all, a synthesis of the most important results of the intellectual structure of the analyzed papers is presented. The intellectual structure is analyzed through the co-citation networks. More in detail, the sources networks and the authors networks are presented in the following.

Figure 3 shows the sources network, i.e., the network among the journals in which the cited papers are published. This figure shows the existence of two main citation groups: (i) a smaller one (colored in red), mainly formed by non-accounting journals (such as Climatic Change, Energy Policy, Ecological Economics, Journal of Cleaner Production etc.) and (ii) a bigger one (colored in blue), mainly formed by typical accounting journals (such as Accounting, Organizations, and Society; Accounting Forum; European Accounting Review, etc.), but also by the Journal of Business Ethics, which is not a typical accounting journal, but which publishes a lot of accounting papers.

Figure 4 shows the authors network, i.e., the network of citations among different authors. Also, in this case they merged two citation groups, but there was no evident distinction between accounting and non-accounting authors. In other words, in each of the two citation groups there are accounting and non-accounting scholars. In both groups, approximately half of the authors are not present in the sample. This means that many authors of the sample are not among the most cited authors in papers dealing with climate change accounting-related topics, based upon insights obtained from the citations in the 85 articles selected for this research.

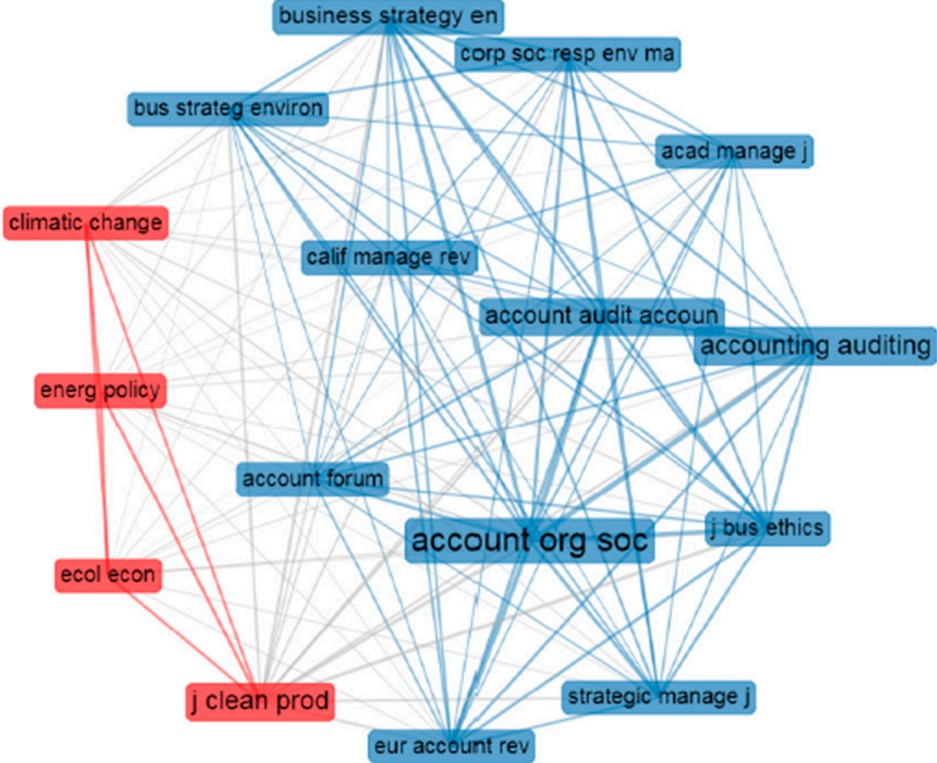

**Figure 3.** Co-citation network: sources.

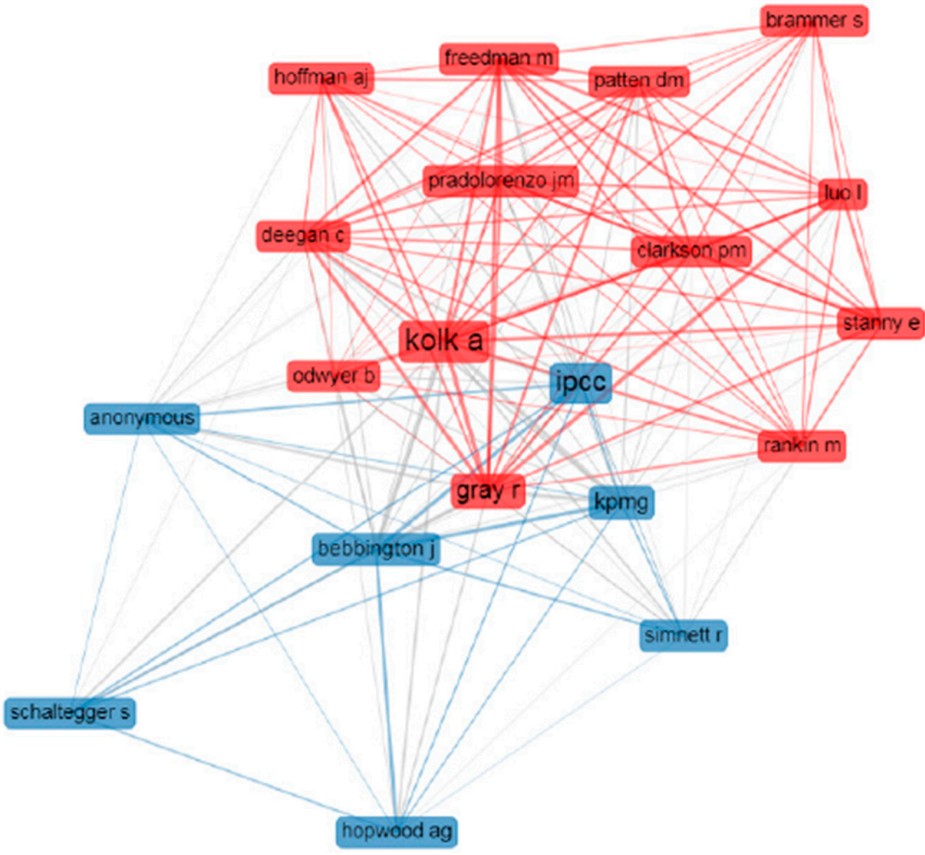

**Figure 4.** Co-citation network: authors.

Figure 5 highlights important aspects about the conceptual structure of the analyzed articles. It shows a thematic map based on the titles of the papers. In the upper part of the matrix, on the left side, the highly developed and isolated themes (niches) are presented. Here we find: "corporate" and "case". Titles dealing with corporate issues and case studies lie in this section of the matrix.

On the right side, the motor themes are presented. Here we find: "assessment" and "change". Titles dealing with emissions assessment and companies' change-related problems fall in this section of the matrix.

Down at the right side, basic and transversal themes are presented: "reporting" and "accounting". The qualitative analysis described in Section 4.5 focuses mainly on these aspects. They are important themes for accounting scholars. In fact, many of the papers falling within the sample deal with different types of sustainability accounting and reporting (e.g., carbon accounting, GHG reporting etc.).

At the left side, emerging or declining themes are presented: "climate", "companies", and "environmental".

Some concepts lie within two areas of the matrix: carbon lies between niche and motor themes, while greenhouse lies between motor and basic/transversal themes. This figure is particularly important for the analysis; in fact, it shows the concept of "climate" within the emerging or declining themes. The authors deemed it is an emerging theme within the sustainability accounting literature, since accounting scholars have only recently started to deal with this theme. More precisely, the analysis conducted in this paper shows that accounting scholars have been dealing with issues related to climate change for some years (e.g., carbon accounting and GHG reporting, mentioned above). However, the terms "climate change accounting/reporting" were not yet particularly widespread in the articles analyzed. This indicates that accounting scholars are progressively moving closer to the issue of climate change. Many articles within the sample dealt with types of accounting and reporting mainly focused on individual aspects of climate change, rather than on the phenomenon as a whole.

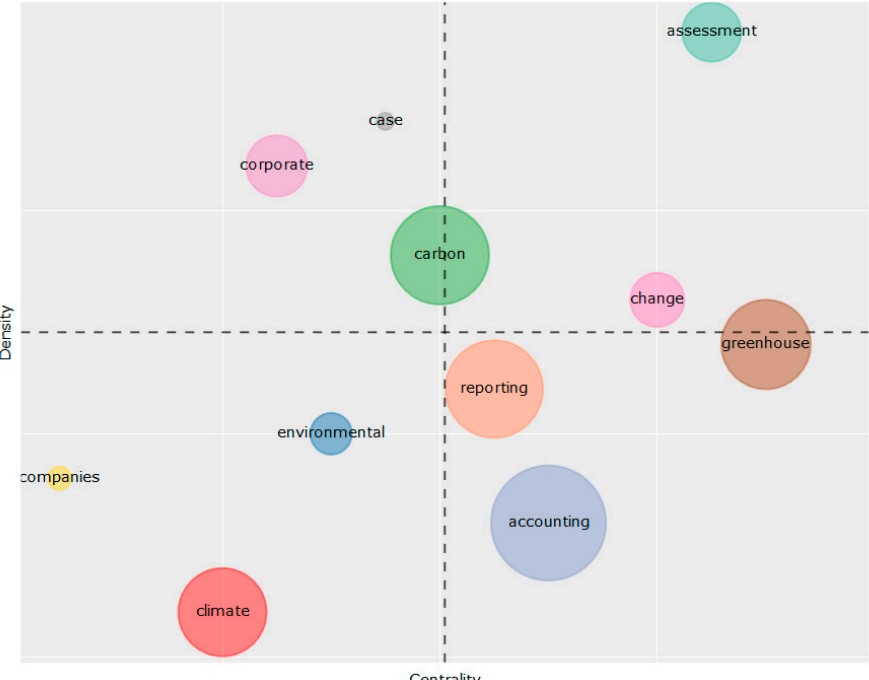

**Figure 5.** Thematic map of the titles of the papers.

This selection of the main results of the network analyses ends with a focus on the social structure, which can be observed through the collaboration networks. The latter can be analyzed observing the following parameters: (a) authors; (b) institutions; (c) countries. No important results emerged for authors' and institutions' collaboration networks. More significant results emerged for the collaboration

networks of countries. Here, there were only a few, small collaboration networks (i.e., among the UK, Canada, the USA, and Sweden and one among Australia, the Netherlands, and New Zealand. Other countries seem to be quite isolated in their climate change accounting-related research.

### 4.5. Qualitative Analysis

In order to answer the RQs, it was essential to also conduct a qualitative analysis, including: (a) the Guthrie et al. [71] analytical framework (and its improved version proposed by Dumay et al. [72]); (b) the scientific background of the most relevant authors of the sample of articles selected for this research.

The results of the Guthrie et al.'s analytical framework were synthesized in Table 7.

**Table 7.** The qualitative framework.

| A. Jurisdiction | Articles | % |
|---|---|---|
| A1. Supranational/international comparative | 4 | 4.71% |
| A2. National | 6 | 7.06% |
| A3. Local government | 6 | 7.06% |
| A4. Public Business Enterprise | 3 | 3.53% |
| A5. Private company | 25 | 29.41% |
| A6. Specific economic sector | 12 | 14.12% |
| A7. One organization | 0 | 0.00 |
| A8. Other | 29 | 34.12% |
| **Tot.** | **85** | **100%** |

| B. Location | Articles | % |
|---|---|---|
| B1. Europe | 19 | 22.35% |
| B2. North America; | 5 | 5.88% |
| B3. South America; | 3 | 3.52% |
| B4. Australasia; | 4 | 4.70% |
| B5 Asia/China; | 8 | 9.41% |
| B6. Africa; | 0 | 0 |
| B7. Other. | 46 | 54.11% |
| **Tot.** | **85** | **100%** |

| C. Focus of the Article | Articles | % |
|---|---|---|
| C1. External reporting | 22 | 25.88% |
| C2. Auditing | 0 | 0.00% |
| C3. Accounting | 23 | 27.06% |
| C4. Governance | 0 | 0.00% |
| C5. Management control/strategy | 8 | 9.41% |
| C6. Performance measurement | 6 | 7,06% |
| C7. Budgeting | 8 | 9.41% |
| C8. Other | 18 | 21.18% |
| **Tot.** | **85** | **100%** |

| D. Research Method | Articles | % |
|---|---|---|
| D1. Case/field study/interviews; | 38 | 44.70% |
| D2. Content analysis/historical analysis; | 7 | 8.23% |
| D3. Survey/questionnaire/other empirical; | 12 | 14.11% |
| D4. Commentary/normative/policy; | 22 | 25.88% |
| D5. Theoretical: literature review/empirical | 6 | 7.05% |
| **Tot.** | **85** | **100%** |

| E. Academics/Practitioners | Articles | % |
|---|---|---|
| E1. Only academic(s) | 69 | 81.17% |
| E2. Only practitioner(s) | 11 | 12.94% |
| E3. Academic(s) and practitioner(s) | 5 | 5.88% |
| **Tot.** | **85** | **100%** |

The results of the qualitative systematic review could be synthesized according to the five categories of the research framework.

As for the jurisdiction (A), in most cases, this criterion was not applicable since there was no specific reference of the articles examined to one of the contexts included in the framework applied. When applicable, the most frequent jurisdiction was a private company (29.41%), followed by specific economic sectors (14.12%). As for the location (B), when applicable, a wide variety of locations was found; the most frequent location was Europe (22.35%), followed by Asia/China (9.41%). As for the focus of the articles (C), in 23 cases the articles were on accounting and 22 on external reporting. There were no articles fully focusing on auditing and governance in the literature analyzed, but there were articles dealing with auditing and governance together with others among the issues categorized in the C dimension of the framework.

It is useful to highlight which are the main aspects addressed within the two most debated focuses identified within dimension C, i.e., accounting and external reporting. As for accounting, most articles focused on carbon and GHG accounting. The papers focused on carbon accounting dealt with different aspects, e.g., the meaning of the concept of carbon accounting, the challenges for research on carbon accounting in management control and performance measurement, comparisons among different forms of carbon and GHG accounting, the need of climate accounting principles, specific applications of carbon footprint accounting in public and private organizations analyzed through cases studies in different locations.

As for papers focusing on external reporting, the papers within the sample dealt with various aspects, e.g., the influence of stakeholders' pressures on GHG emissions reporting, the characteristics of corporate GHG emissions reporting, explorations of the current status and key determinants of corporate disclosure on climate change, the role of GHG reporting in impression management and legitimacy issues, impacts of carbon performance on climate change disclosure, relationships between board effectiveness and voluntary disclosure of climate change information, effects of GHG emissions on firms' performances, climate change performance measurement, comparisons of GHG reporting methodologies, voluntary and mandatory climate change reporting, national GHG inventories, factors affecting GHG emissions reporting, climate aspects in sustainability reporting.

As for the research methods (D), in most cases the articles analyzed fell within the categories "case/field study/interview" or "commentary/normative/policy". As for the academic or practitioner authors (F), in most cases, the articles analyzed fell within the category "academic(s)".

From this systematic literature review it emerged that the current state of the literature on climate change accounting is not well developed, especially on the part of the practitioners, which means that professional standards for this field have not yet been identified.

The qualitative analysis included the examination of the scientific background of the most important authors of the 85 articles selected for this research. The analysis was based on the academic and professional roles of the highly productive authors of the selected articles to understand whether they can be considered as "accounting scholars" or not. In this context, an analysis was conducted on the authors (see Table 3), focused on the 15 authors who published at least two articles within the sample. The analysis was performed by evaluating information about these authors available on the web and dividing them into:

- No. 9 authors classifiable as "accounting scholars", since their (academic or professional) role, research interests, research areas, and scientific/professional background clearly referred to accounting-related subjects;
- No. 6 authors not classifiable as "accounting scholars", since no clear references to accounting-related subjects were found in their curricula. In most cases, their research interests referred to "carbon accounting" or "GHG accounting", but their scientific background and/or their academic/professional roles were not related to typical accounting subjects. Some of them are scholars of atmospheric physics, environmental biology, climate change policy, and politics.

This simple analysis helped the authors of this article to develop a rudimentary picture of the scientific backgrounds of the most productive authors of articles among the 85 selected for this research. This analysis revealed a fragmented situation, in which not only typical "accounting scholars" write about climate change accounting-related topics, but also scholars of disciplines different from accounting, who embrace climate change accounting-related topics only in a marginal way. This is not surprising considering the multidisciplinary nature of climate change accounting/reporting/disclosure and owing to the rapidly evolving borders of scientific disciplines (in general) and of the accounting discipline (in particular). These aspects were already highlighted in Sections 1 and 2 of this paper, as the authors reflected upon the lack of clear margins of sustainability accounting and accountability in general and of climate change-related accounting and accountability in particular.

It was noteworthy that some "accounting scholars" published articles within 85 articles selected for this research as co-authors with "non-accounting scholars". This is an encouraging sign of a multidisciplinary approach among the authors in collaboration among scholars of different research areas.

More in-depth considerations of these aspects are presented in Section 5, which is devoted to a discussion about the current status and the future trajectories of climate change-related accounting and accountability literature and which provides suggestions for future developments of climate change-related accounting and accountability research.

## 5. Discussion

As stated in the previous sections, the margins of scientific disciplines are not fixed; they are flexible and depend upon geographical, cultural, and temporal variables [74]. This is also true for accounting and for its subarticulations, such as environmental aspects, which started to be considered by accounting scholars in the 1960s and 1970s, but has expanded rapidly since the late 1980s and early 1990s. The attention towards climate change accounting-related aspects developed at the beginning of the 21st century [75]. However, climate change accounting-related literature is not very widespread [37].

This bibliometric analysis on climate change accounting and reporting-related literature allowed the authors to answer the research questions stated in Section 1.

RQ1 asked "What is the knowledge base of climate change accounting/reporting and its intellectual structure?" According to bibliometrix, the knowledge base of a research field is analyzed through a complex series of information relating to the following aspects: dataset, sources (in our case, the journals in which the analyzed papers were published), authors of the analyzed papers, and documents (i.e., articles analyzed, cited references, keywords, and recurring concepts in the titles and abstracts).

As for the dataset, Section 4.1 displayed the main information about the selected articles in Table 4. Significant trends were observed in publications about this topic in the analyzed time span (Figure 1)—starting in 2008 there was a considerable growth of articles identified in this literature review.

The results of this review show that there is scope for accounting to support a move towards climate change issues. Climate change adaptation is necessary for companies, both for business strategy and for risk management [17]. Climate change impacts on infrastructure and supply chains. Reputational, legal, and regulatory obligations emerge, and consequently decision-makers need information to support the valuation of the economic implications of climate impacts. They need information to identify risks and liabilities, to effect cost–benefit analyses, and innovate performance and benchmarking metrics [24]. To provide answers to these questions, the role of managerial accounting is very important.

The role of managerial accounting and financial accounting is crucial in climate risk evaluation and in value creation. GHG emissions and carbon accounting produce information about the costs and benefits of mitigation strategies (e.g., emission reductions implying energy savings). Particular challenges for accountants are created by emission rights. This information is voluntarily disclosed and supports stakeholders' decisions and organizational policy making.

RQ2 asked "What is the research front (or conceptual structure) of this research field?" According to bibliometrix, the conceptual structure of a research field is analyzed through information relating to the following main aspects: co-occurrence networks, thematic maps, and factorial analyses. For synthesis

needs, this paper presents only a thematic map of the titles of the papers, aiming at describing the categorization of the words within the titles according to their state of development within the literature (Figure 5). Bibliometrix allows also thematic maps of keywords plus, authors' keywords, and abstracts to be obtained (whose figures have not been included in this paper). The most relevant information that was extracted from the thematic map of titles, which revealed that:

1. The concepts of "assessment" and "change" were the main motor themes;
2. The concept of "carbon" was within niche themes and motor themes, while "greenhouse" lay between motor and basic/transversal themes;
3. The concepts of "reporting" and "accounting" were the main basic and transversal themes;
4. The concepts of "climate", "environmental", and "companies" were the main emerging or declining themes;
5. The concepts of "corporate" and "case" were the main highly developed and isolated themes.

Analyzing also the thematic maps of keywords and abstracts (whose figures have not been included in this paper, as stated above), the word "climate" was within the emerging themes and "climate-change" was within the niche themes. "Accounting" and "reporting" were the main basic and transversal themes, while "carbon" and "greenhouse" were between the highly developed and the motor themes. This seems to confirm that "accounting" and "reporting" about carbon and other greenhouse emissions are the main topics addressed in the analyzed literature, while articles explicitly devoted to "climate" and "climate-change" accounting and reporting are a marginal portion of this literature.

These results confirmed the initial state of development of climate change accounting-related literature and open the way to a series of reflections on the opportunities for accounting scholars to focus their future research on this issue, the importance of which for all society is very high and urgently needed.

The authors of this paper are convinced that the development of climate change accounting/reporting-related literature will follow a path similar to the one that has shaped social, environmental, and sustainability accounting. According to Gray [14], although this field of research "may well have started his life as an (inter-disciplinary) element within accounting [ . . . ] it is now emerging as a trans-disciplinary field in its own right; drawing heavily on accounting but no longer bound by it". As stated by Gray [14] for social, environmental, and sustainability accounting, the authors of this paper are convinced that climate change accounting/accountability "is too important to lapse back into [ . . . ] the predictability of normal science: the issues are much too crucial".

RQ3 asked "What is the social network structure of the scientific community writing about climate change accounting/reporting?" Bibliometrix analyzes the social network structure of a research field through the collaboration networks, which can involve: (i) single authors, (ii) single institutions (i.e., the authors' affiliations), and (iii) countries. Given the scarce significance of the results about points (i) and (ii), the authors of this paper presented information only about the main collaboration networks among countries. It was found to be highly fragmented, with a few small collaboration networks and a predominance of countries that are isolated in their climate change accounting-related research. This is another element confirming the relative newness of this research field and there is a great opportunity to promote intra- and extra-disciplinary collaborations among scholars of different geographical origins.

RQ4 asked: "What lessons can be learned from the analyzed literature about the current state and the future developments of the climate change accounting and reporting-related literature?" The answers to this question are addressed in Section 6 (conclusions) as to the current state of the climate change accounting and reporting-literature and in Section 7 (future research directions) as to future developments.

## 6. Conclusions

According to Doni [76], the connection between the climate change regime and sustainable development is clear: renewable energy can improve energy access and health benefits related to the reduction of polluting emissions; actions against climate change can also improve social issues such as fragility, displacement, migration, and conflict. Furthermore, climate change allows obtaining a successful implementation of all 17 SDGs. The impact of climate change accounting on SDG no. 13 targets and indicators is strong. Specifically, this literature review shows that the link arises in terms of information and data that accounting provides ($CO_2$ and greenhouse gas emissions) to implement the adaptation and mitigation strategies of firms; in fact, the main variables that can be subjected to managerial control are the $CO_2$ and greenhouse gas emissions.

The main perspectives addressed by climate change accounting-related research were external reporting and accounting, while less attention was devoted to auditing, governance, management control/strategy, and performance measurement. These results provide a broader evidence than previous studies about the topics addressed by accounting scholars when dealing with sustainability (and, more specifically, with climate change) topics. In fact, the few studies analyzing accounting publications about the SDGs show that SDG no. 13 is one of the most analyzed sustainable development goals in the academic literature, but they do not focus on the different accounting research areas falling within this topic.

The theoretical and practical implications of this research underline the need for a deeper involvement of accounting scholars and practitioners in SDG accounting and reporting and, more in detail, in climate change accounting and reporting. Although publications on this research area have been increasing in recent years, there is still a need for an increase in the interest of accounting scholars and practitioners in this area. In fact, the information needs of corporate stakeholders are moving towards a greater and better knowledge of what companies are doing to contribute to a more sustainable future for all.

The first part of RQ4 focused on the current state of climate change accounting and reporting-related literature. The literature review here performed allowed us to identify the main topics addressed in the analyzed papers. They mainly related to sustainability-related accounting and reporting. Less attention was devoted to auditing, management control/strategy, and performance measurement which are, indeed, relevant topics to be addressed. Few of the articles within the sample dealt with sustainability accounting and reporting "in the strict sense" (i.e., explicitly mentioning "sustainability accounting" or "sustainability reporting" in their title or abstract). Most of the articles analyzed dealt with sustainability accounting and reporting "in a broad sense" (i.e., without mentioning "sustainability accounting" or "sustainability reporting" in their title or abstract, but dealing with sustainability and climate change-related accounting and reporting, e.g., carbon accounting, GHG accounting and reporting, footprint reporting, etc.). Many papers focused on the current state and future perspectives of research on this issue. It is therefore clear that accounting scholars are questioning their role in the context of sustainability and the struggle against climate change. They are also questioning the role they should have in order to make a useful contribution in this regard.

## 7. Future Research Directions

The main limitation of this paper can be synthesized as follows. The authors only searched for climate change accounting and reporting publications in the dataset of WoS, which could potentially ignore other publications. However, since WoS covers a wide range of leading accounting journals, we believe we have identified most of the relevant articles on the matter. Responsibility for any errors or omissions lies with the authors.

In conclusion, the authors of this paper encourage the accounting scientific community to engage more with climate change accounting and reporting research. In fact, there are several less explored or unexplored research opportunities related to climate change accounting and reporting. A deeper involvement of accounting scholars in this research area could be beneficial for both the stakeholders of companies and citizens.

The authors deem that the main gaps to be filled in this literature are:

- A greater concentration should be invested on the concrete effects produced by climate change-related practices on the sustainability performances of firms;
- A broader focus of accounting scholars on all the 17 SDGs instead of one part of them (e.g., carbon emissions, other GHG emissions, etc.) would be useful;
- A greater attention toward climate- elated risk management and its concrete effects on the risks of firms is urgently needed;
- A broader attention should be given to climate change management practices (including management accounting), which seems, at the moment, to be of less importance in the climate change-related accounting literature in comparison with reporting-related aspects. In fact, reporting could be driven by various motivations (e.g., greenwashing, window dressing, industry/legislative/stakeholder pressures, etc.) [47], not always coherent with the real achievement of the SDGs. A greater focus of the research community on management decisions and actions to counteract climate change, would be helpful to promote the understanding and development of such practices. The analyzed literature seemed, on the contrary, to focus mainly on reporting-related aspects (as shown by the great number of articles about sustainability reporting, both in the strict and in the broad sense). Even if disclosure is an important aspect to focus upon, it is not the main problem. The authors deem that the most urgent problem to be addressed is the management of climate change-related aspects, with specific reference to strategic and operational planning, accounting, and control of the actions implemented by the management of firms to counter climate change problems. These considerations are similar to those proposed by Gibson [77], as she stated that "reporting [ … ] is not the real problem in addressing atmospheric pollution, but that the economic philosophy which attempts to address ecological problems in economic terms is the main culprit".

These gaps should become the main areas in which accounting scholars should focus during the next decade in their sustainability (and more specifically in their climate change-related) research.

As already stated, the perspectives of development of the accounting discipline in climate change-related aspects are very interesting. In fact, the matter is too important to fade into oblivion!

Considering that accounting is only one (and not necessarily the most important) discipline involved in climate change-related problems, the authors of this paper suggest that the future development of the accounting discipline will include:

- A growth of publications about climate change-related aspects, not only in typical academic accounting journals, but also in more interdisciplinary journals;
- A greater collaboration among authors, both within the accounting discipline and with scholars of other scientific areas;
- A greater collaboration between academic scholars, practitioners, and policy makers.

In this context, accounting scholars could make highly important contributions about:

- The implementation and modification of existing and creation of new managerial tools to counteract the impacts of firms on sustainability and climate change;
- The study of effective results of managerial actions designed to help companies to dramatically reduce their impact on unsustainability and climate change;
- Analyses of business cases (e.g., case studies, field studies, single best practices, benchmarking among best practices, etc.) that could help the managers of firms to become involved and committed to engaging their colleagues, employees, and customers in achieving sustainability-related objectives;
- Stimulating raising awareness among business managers, other accounting scholars, and students about the needs to address climate change-related problems, not only for opportunistic reasons (e.g., window dressing), but also because this is one of the main problems that individuals, public institutions, and companies (both large corporations and small businesses) should face now to achieve a better and more sustainable future for everyone.

**Author Contributions:** Conceptualization, authors C.G. and P.P.; methodology, authors C.G. and P.P.; software, author C.G.; validation, authors C.G., P.P., and V.L.; formal analysis, author C.G.; investigation, authors C.G. and V.L.; resources, author C.G.; data curation, authors C.G. and V.L.; writing—original draft preparation, authors P.P. and C.G.; writing—review and editing, author D.H.; visualization, author P.P.; supervision, author D.H.; project administration, author C.G. The idea of the paper and the bibliographical researches are due to the common work of the authors. Sections 1, 2 and 6 are to be attributed to P.P.; Sections 3, 3.2, 3.3, 4, 4.1, 4.2 and 5 are to be attributed to C.G.; Sections 3.1 and 4.3–4.5 are to be attributed to V.L.; Section 7 is to be attributed to D.H. All authors have read and agreed to the published version of the manuscript.

**Funding:** This research received no external funding.

**Conflicts of Interest:** The authors declare no conflict of interest.

## Appendix A. The 85 Articles Analyzed

1.  Ajani, J.I.; Keith, H.; Blakers, M.; Mackey, B.G.; King, H.P. Comprehensive carbon stock and flow accounting: A national framework to support climate change mitigation policy. *Ecol. Econ.* **2013**, *89*, 61–72.

2.  Andrew, J.; Cortese, C. Free market environmentalism and the neoliberal project: The case of the climate disclosure standards board. *Crit. Perspect. Account.* **2013**, *24*, 397–409.

3.  Ascui, F.; Lovell, H. As frames collide: Making sense of carbon accounting. *Account. Audit. Account. J.* **2011**, *24*.

4.  Ascui, F.; Lovell, H. Carbon accounting and the construction of competence. *J. Clean. Prod.* **2012**, *36*, 48–59.

5.  Atkins, J.; Atkins, B.C.; Thomson, Ian; Maroun, W. "Good" news from nowhere: Imagining utopian sustainable accounting. *Account. Audit. Account. J.* **2015**, *28*.

6.  Barrett, J.; Peters, G.; Wiedmann, T.; Scott, K.; Lenzen, M.; Roelich, K.; Le Quere, C. Consumption-based ghg emission accounting: A UK case study. *Clim. Policy* **2013**, *13*, 451–470.

7.  Bebbington, J.; Larrinaga-Gonzalez, C. Carbon trading: Accounting and reporting issues. *Eur. Account. Rev.* **2008**, *17*, 697–717.

8.  Ben-Amar, W.; Chang, M.; McIlkenny, P. Board gender diversity and corporate response to sustainability initiatives: Evidence from the carbon disclosure project. *J. Bus. Ethics* **2017**, *142*, 369–383.

9.  Ben-Amar, W.; McIlkenny, P. Board effectiveness and the voluntary disclosure of climate change information. *Bus. Strategy Environ.* **2015**, *24*, 704–719.

10. Birdsey, R.A. Carbon accounting rules and guidelines for the United States forest sector. *J. Environ. Qual.* **2006**, *35*, 1518–1524.

11. Boston, J.; Lempp, F. Climate change explaining and solving the mismatch between scientific urgency and political inertia. *Account. Audit. Account. J.* **2011**, *24*, 1000–1021.

12. Brandao, M.; Levasseur, A.; Kirschbaum, M.U.F.; Weidema, B.P.; Cowie, A.L.; Jorgensen, S.V.; Hauschild, M.Z.; Pennington, D.W.; Chomkhamsri, K. Key issues and options in accounting for carbon sequestration and temporary storage in life cycle assessment and carbon footprinting. *Int. J. Life Cycle Assess.* **2013**, *18*, 230–240.

13. Brander, M. Comparative analysis of attributional corporate greenhouse gas accounting, consequential life cycle assessment, and project/policy level accounting: A bioenergy case study. *J. Clean. Prod.* **2017**, *167*, 1401–1414.

14. Brander, M. Transposing lessons between different forms of consequential greenhouse gas accounting: Lessons for consequential life cycle assessment, project-level accounting, and policy-level accounting. *J. Clean. Prod.* **2016**, *112*, 4247–4256.

15. Brouhle, K.; Harrington, D.R. Firm strategy and the Canadian voluntary climate challenge and registry (vcr). *Bus. Strategy Environ.* **2009**, *18*, 360–379.

16. Burritt, R.L.; Schaltegger, S.; Zvezdov, D. Carbon management accounting: Explaining practice in leading German companies. *Aust. Account. Rev.* **2011**, *21*, 80–98.

17. Burritt, R.L.; Tingey-Holyoak, J. Forging cleaner production: The importance of academic-practitioner links for successful sustainability embedded carbon accounting. *J. Clean. Prod.* **2012**, *36*, 39–47.

18. Bustamante, M.C.; Silva, J.S.O.; Cantinho, R.Z.; Shimbo, J.Z.; Oliveira, P.V.C.; Santos, M.M.O.; Ometto, J.P.H.B.; Cruz, M.R.; Mello, T.R.B.; Godiva, D.; et al. Engagement of scientific community and transparency in C accounting: The Brazilian case for anthropogenic greenhouse gas emissions from land use, land-use change and forestry. *Environ. Res. Lett.* **2018**, *13*, 55005.

19. Byrne, Susan; O'Regan, B. Material flow accounting for an Irish rural community engaged in energy efficiency and renewable energy generation. *J. Clean. Prod.* **2016**, *127*, 363–373.

20. Comyns, B. Determinants of ghg reporting: An analysis of global oil and gas companies. *J. Bus. Ethics* **2016**, *136*, 349–369.

21. Cooper, S.; Pearce, G. Climate change performance measurement, control and accountability in English local authority areas. *Account. Audit. Account. J.* **2011**, *24*.

22. Cooper, S.A.; Raman, K.; Yin, J. Halo effect or fallen angel effect? Firm value consequences of greenhouse gas emissions and reputation for corporate social responsibility. *J. Account. Public Policy* **2018**, *37*, 226–240.

23. Cordova, C.; Zorio-Grima, A.; Merello, P. Carbon emissions by South American companies: Driving factors for reporting decisions and emissions reduction. *Sustainability* **2018**, *10*, 2411.

24. Cowie, A.L.; Kirschbaum, M.U.F.; Ward, M. Options for including all lands in a future greenhouse gas accounting framework. *Environ. Sci. Policy* **2007**, *10*, 306–321.

25. Csutora, M.; Mozner, Z.V. Proposing a beneficiary-based shared responsibility approach for calculating national carbon accounts during the post-Kyoto era. *Clim. Policy* **2014**, *14*, 599–616.

26. Deckmyn, G.; Muys, B.B.; Quijano, J.G.; Ceulemans, R. Carbon sequestration following afforestation of agricultural soils: Comparing oak/beech forest to short-rotation poplar coppice combining a process and a carbon accounting model. *Glob. Chang. Biol.* **2004**, *10*, 1482–1491.

27. Depoers, F.; Jeanjean, T.s; Tiphaine, J. Voluntary disclosure of greenhouse gas emissions: Contrasting the carbon disclosure project and corporate reports. *J. Bus. Ethics* **2016**, *134*, 445–461.

28. Dilling, P.F.A.; Harris, P. Reporting on long-term value creation by Canadian companies: A longitudinal assessment. *J. Clean. Prod.* **2018**, *191*, 350–360.

29. Ellison, D.; Lundblad, M.; Petersson, H. Carbon accounting and the climate politics of forestry. *Environ. Sci. Policy* **2011**, *14*, 1062–1078.

30. Gallego, A.I. Impact of co2 emission variation on firm performance. *Bus. Strategy Environ.* **2012**, *21*, 435–454.

31. Gallego-Alvarez, I.; Garcia-Sanchez, I.M.; Da Silva, V.C. Climate change and financial performance in times of crisis. *Bus. Strategy Environ.* **2014**, *6*, 361–374.

32. Gentil, E.; Christensen, T.H.; Aoustin, E. Greenhouse gas accounting and waste management. *Waste Manag. Res.* **2009**, *27*, 696–706.

33. Gerst, M.D.; Howarth, R.B.; Borsuk, M.E. Accounting for the risk of extreme outcomes in an integrated assessment of climate change. *Energy Policy* **2010**, *38*, 4540–4548.

34. Giannarakis, G.; Zafeiriou, E.; Sariannidis, N. The impact of carbon performance on climate change disclosure. *Bus. Strategy Environ.* **2017**, *26*, 1078–1094.

35. Gillenwater, M. Forgotten carbon: Indirect co2 in greenhouse gas emission inventories. *Environ. Sci. Policy* **2008**, *11*, 195–203.

36. Gillett, N.P.; Matthews, H.D. Accounting for carbon cycle feedbacks in a comparison of the global warming effects of greenhouse gases. *Environ. Res. Lett.* **2010**, *5*, 34011.

37. Grauel, J.; Gotthardt, D. The relevance of national contexts for carbon disclosure decisions of stock-listed companies: A multilevel analysis. *J. Clean. Prod.* **2016**, *133*, 1204–1217.

38. Gustavsson, L.; Karjalainen, T.; Marland, G.; Savolainen, I.; Schlamadinger, B.; Apps, M. Project-based greenhouse-gas accounting: Guiding principles with a focus on baselines and additionality. *Energy Policy* **2000**, *28*, 935–946.

39. Gusti, M.; Jonas, M. Terrestrial full carbon account for Russia: Revised uncertainty estimates and their role in a bottom-up/top-down accounting exercise. *Clim. Chang.* **2010**, *103*, 159–174.

40. Haddock-Fraser, J.E.; Tourelle, M. Corporate motivations for environmental sustainable development: Exploring the role of consumers in stakeholder engagement. *Bus. Strategy Environ.* **2010**, 19, 527–542.

41. Halkos, G.; Skouloudis, A. Exploring the current status and key determinants of corporate disclosure on climate change: Evidence from the Greek business sector. *Environ. Sci. Policy* **2016**, *56*, 22–31.

42. Hao, Y.; Su, M.; Zhang, L.; Cai, Y.; Yang, Z. Integrated accounting of urban carbon cycle in Guangyuan, a mountainous city of china: The impacts of earthquake and reconstruction. *J. Clean. Prod.* **2015**, *103*, 231–240.

43. Haque, S.; Deegan, C. Corporate climate change-related governance practices and related disclosures: Evidence from Australia. *Aust. Account. Rev.* **2010**, *20*, 313–333.

44. Haripriya, G.S. Carbon budget of the Indian forest ecosystem. *Clim. Chang.* **2003**, *56*, 291–319.

45. Hartmann, Frank; Perego, P.; Young, A. Carbon accounting: Challenges for research in management control and performance measurement. *Abacus A J. Account. Financ. Bus. Stud.* **2013**, *49*, 539–563.

46. Hasselmann, K. Intertemporal accounting of climate change—Harmonizing economic efficiency and climate stewardship. *Clim. Chang.* **1999**, *41*, 333–350.

47. Hoerisch, J.; Ortas, E.; Schaltegger, S.; Alvarez, I. Environmental effects of sustainability management tools: An empirical analysis of large companies. *Ecol. Econ.* **2015**, *120*, 241–249.

48. Jasinevicius, G.; Lindner, M.; Cienciala, E.; Tykkylainen, M. Carbon accounting in harvested wood products assessment using material flow analysis resulting in larger pools compared to the IPCC default method. *J. Ind. Ecol.* **2018**, *22*, 121–131.

49. Jing, R.; Cheng, J.C.P.; Gan, V.J.L.; Woon, K.S.; Lo, I.M.C. Comparison of greenhouse gas emission accounting methods for steel production in China. *J. Clean. Prod.* **2014**, *83*, 165–172.

50. Kalu, J.U.; Buang, A.; Aliagha, G.U. Determinants of voluntary carbon disclosure in the corporate real estate sector of Malaysia. *J. Environ. Manag.* **2016**, *182*, 519–524.

51. Kolk, A.; Levy, D.; Pinkse, J. Corporate responses in an emerging climate regime: The institutionalization and commensuration of carbon disclosure. *Eur. Account. Rev.* **2008**, *17*, 719–745.

52. Larsen, H.N.; Hertwich, E.G. Implementing carbon-footprint-based calculation tools in municipal greenhouse gas inventories. *J. Ind. Ecol.* **2010**, *14*, 965–977.

53. Lee, K. Carbon accounting for supply chain management in the automobile industry. *J. Clean. Prod.* **2012**, *36*, 83–93.

54. Liesen, A.; Hoepner, A.G.; Patten, D.M.; Figge, F. Does stakeholder pressure influence corporate ghg emissions reporting? Empirical evidence from Europe. *Account. Audit. Account. J.* **2015**, *28*.

55. Liptow, C.; Janssen, M.; Tillman, A. Accounting for effects of carbon flows in LCA of biomass-based products exploration and evaluation of a selection of existing methods. *Int. J. Life Cycle Assess.* **2018**, *23*, 2110–2125.

56. Lodhia, S.; Martin, N. Stakeholder responses to the national greenhouse and energy reporting act an agenda setting perspective. *Account. Audit. Account. J.* **2012**, *25*.

57. Lozano, R.; Huisingh, D. Inter-linking issues and dimensions in sustainability reporting. *J. Clean. Prod.* **2011**, *19*, 99–107.

58. Martire, S.; Mirabella, N.; Sala, S. Widening the perspective in greenhouse gas emissions accounting: The way forward for supporting climate and energy policies at municipal level. *J. Clean. Prod.* **2018**, *176*, 842–851.

59. Milne, M.J.; Grubnic, S. Climate change accounting research: Keeping it interesting and different. *Account. Audit. Account. J.* **2011**, *24*.

60. Mjelde, A.; Martinsen, K.; Eide, M.; Endresen, O. Environmental accounting for arctic shipping—A framework building on ship tracking data from satellites. *Mar. Pollut. Bull.* **2014**, *87*, 22–28.

61.  Mozner, Z.V. A consumption-based approach to carbon emission accounting—Sectoral differences and environmental benefits. *J. Clean. Prod.* **2013**, *42*, 83–95.

62.  Neelis, M.L.; Patel, M.; Gielen, D.J.; Blok, K. Modelling co2 emissions from non-energy use with the non-energy use emission accounting tables (neat) model. *Resour. Conserv. Recycl.* **2005**, *45*, 226–250.

63.  Pearson, T.R.H.; Brown, S.; Andrasko, K. Comparison of registry methodologies for reporting carbon benefits for afforestation projects in the United States. *Environ. Sci. Policy* **2008**, *11*, 490–504.

64.  Pellegrino, C.; Lodhia, S. Climate change accounting and the Australian mining industry: Exploring the links between corporate disclosure and the generation of legitimacy. *J. Clean. Prod.* **2012**, *36*, 68–82.

65.  Prado-Lorenzo, J.; Rodriguez-Dominguez, L.; Gallego-Alvarez, I.; Garcia-Sanchez, I. Factors influencing the disclosure of greenhouse gas emissions in companies world-wide. *Manag. Decis.* **2009**, *47*.

66.  Pulles, T.; Yang, H. Ghg emission estimates for road transport in national ghg inventories. *Clim. Policy* **2011**, *11*, 944–957.

67.  Radhouane, I.; Nekhili, M.; Nagati, H.; Pache, G. The impact of corporate environmental reporting on customer-related performance and market value. *Manag. Decis.* **2018**, *56*.

68.  Rankin, M.; Windsor, C.; Wahyuni, D. An investigation of voluntary corporate greenhouse gas emissions reporting in a market governance system Australian evidence. *Account. Audit. Account. J.* **2011**, *24*.

69.  Rickels, W.; Rehdanz, K.; Oschlies, A. Methods for greenhouse gas offset accounting: A case study of ocean iron fertilization. *Ecol. Econ.* **2010**, *69*, 2495–2509.

70.  Rogelj, J.; Meinshausen, M.; Schaeffer, M.; Knutti, R.; Riahi, K. Impact of short-lived non-co2 mitigation on carbon budgets for stabilizing global warming. *Environ. Res. Lett.* **2015**, *10*, 75001.

71.  Schaltegger, S.; Csutora, M. Carbon accounting for sustainability and management. Status quo and challenges. *J. Clean. Prod.* **2012**, *36*, 1–16.

72.  Shvidenko, A.; Schepaschenko, D.; McCallum, I.; Nilsson, S. Can the uncertainty of full carbon accounting of forest ecosystems be made acceptable to policymakers? *Clim. Chang.* **2010**, *103*, 137–157.

73.  Milne, M.J.; Grubnic, S.; Solomon, J.F.; Solomon, A.; Norton, S.D.; Joseph, N.L. Private climate change reporting: an emerging discourse of risk and opportunity? *Account. Audit. Account. J.* **2011**, *24*.

74.  Steckel, J.C.; Kalkuhl, M.; Marschinski, R. Should carbon-exporting countries strive for consumption-based accounting in a global cap-and-trade regime? *Clim. Chang.* **2010**, *100*, 779–786.

75.  Sullivan, R.; Gouldson, A. Does voluntary carbon reporting meet investors' needs? *J. Clean. Prod.* **2012**, *36*, 60–67.

76.  Talbot, D.; Boiral, O. Ghg reporting and impression management: An assessment of sustainability reports from the energy sector. *J. Bus. Ethics* **2018**, *147*, 367–383.

77.  Tang, S.; Demeritt, D. Climate change and mandatory carbon reporting: Impacts on business process and performance. *Bus. Strategy Environ.* **2018**, *4*, 437–455.

78.  Thomae, J.; Dupre, S.; Hayne, M. A taxonomy of climate accounting principles for financial portfolios. *Sustainability* **2018**, *10*, 328.

79.  Tokarska, K.B.; Gillett, N.P.; Arora, V.K.; Lee, W.G.; Zickfeld, K. The influence of non-co2 forcings on cumulative carbon emissions budgets. *Environ. Res. Lett.* **2018**, *13*, 34039.

80.  Trotman, A.J.; Trotman, K.T. Internal audit's role in ghg emissions and energy reporting: Evidence from audit committees, senior accountants, and internal auditors. *Audit. J. Pract. Theory* **2015**, *34*, 199–230.

81.  Horacio Villarino, S.; Studdert, G.A.; Laterra, P.; Cendoya, M.G. Agricultural impact on soil organic carbon content: Testing the IPCC carbon accounting method for evaluations at county scale. *Agric. Ecosyst. Environ.* **2014**, *185*, 119–132.

82.  Whittaker, C.; McManus, M.C.; Smith, P. A comparison of carbon accounting tools for arable crops in the United Kingdom. *Environ. Model. Softw.* **2013**, *46*, 228–239.

83. Williams, S.; Schaefer, A. Small and medium-sized enterprises and sustainability: managers' values and engagement with environmental and climate change issues. *Bus. Strategy Environ.* **2013**, *22*, 173–186.

84. Wu, L.; Mao, X.Q.; Zeng, A. Carbon footprint accounting in support of city water supply infrastructure siting decision making: A case study in Ningbo, China. *J. Clean. Prod.* **2015**, *103*, 737–746.

85. Zhang, C.; Han, R.; Yu, B.; Wei, Y. Accounting process-related co2 emissions from global cement production under shared socioeconomic pathways. *J. Clean. Prod.* **2018**, *184*, 451–465.

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
