# Peer review of "Climate Change Accounting and Reporting: A Systematic Literature Review"

_sustainability, doi:10.3390/su12135455_

Round 1

Reviewer 1 Report

The paper analyzes an interesting and novel field of study in business and management literature. The research method used by the authors is adequate to the research question. Thus, I think that the paper could represents a relevant contribution within the debate.

However, the paper requires some minor improvements in order to be published.  Those are my suggestions:

  1. The paper could be reinforced through a better connection with prior paper on SDGs. In particular, I have seen that the authors in some parts cited the official communication of the United Nations despite an increasing number of academics have started to discuss about “Sdg accounting”.
  2. The research question is not related to a specific research gap. Thus, the introduction requires further connection with prior studies on climate change and accounting. On the point, the Sustainability, Accounting, Management and Policy Journal published a special issue on carbon accounting.
  3. There are some paragraphs composed by only one sentence (example: page 3, the first two paragraphs of Research Background). Please revise
  4. Why have you used WoS instead of Scopus or EBSCO? Your methodology is correct but it needs a justification.
  5. What are the difference between Table 7 and Figure 4? I think that they provide the same information.
  6. The Table 2, 4 and 5 are not adequate to the editorial standard of Sustainability. Please revise them
  7. The bibliometric outputs are difficult to read due to the low quality of the pictures. Please try to replace them with figures that are more readable.
  8. The conclusion could be extended through a more detailed discussion regard the theoretical contribution provided to the scientific debate. Furthermore, you could divide discussions, future research directions and conclusions in subparagraph in order to favor the comprehension of the main insights collected within the paper.
  9. Please check the references.

In conclusion, I think that the paper could be adequate for the publication after a round of minor revision.

Reviewer 2 Report

The special issue that this article seeks to engage with is very important and the issues the authors seek to contribute to are very important and timely. However, in its current structure, the article does not make such a contribution. It reads as either lists of findings (in bullet points) or short answers to questions. The discussion at the end addresses the research questions listed earlier and answers them in bullet points. A more solid and causality based narrative us necessary.

The analysis of the data is also too descriptive. It references the quantity and topics in the articles analyzed rather than reflecting on the relevance of the literature for the overall topic of accounting and accountability.   There is no clear connection to the climate change regime and to sustainable development. The article is about research agendas in accounting, and it is not clear whether this fits the purpose of the special issue for which this article is intended. Based on the title of the article, and the overall topic of the special issue “Accounting and Accountability for the SDGs”, I was expecting an analysis of what is being done to generate information and accountability that connects efforts on climate change to the indicators in SDG13.    Below are some specific comments:   - The article establishes the idea of accounting, accountability and reporting, but it is only addresses the first issue. Yet, from the perspective of the SDGs the other two are even more relevant. - The structure of the abstract does not explain the importance of the topic. - The article refers to different categories of reporting related to sustainable development but those are not defined or explained. And this makes it more difficult to understand the discussion. - The authors state that sustainable development (as a concept) is the main driver for reporting. This is conflation of concepts and inaccurate characterization.  - There are a number of issues with the methodology. The bibliometrics approach is not explained, and there is no connection between this type of analysis and the SDGs. - Many of the graphs are very difficult to understand. What is their value added? - It is difficult to follow the relation from the data presented and the conclusions. - The categories on reporting presented on the discussion around the (C) variable were not explained at any point. It makes it difficult to understand the meaning of the results.   The paper does not offer a clear contribution to the core issue of the SDGs and fails to explain why an expanded research agenda on climate change accounting would be important for sustainability.

Author Response

Response to Reviewer 3 Comments

Point 1: Extensive editing of English language and style required

Response 1: A native speaker accounting expert has checked and corrected English language and style.

Point 2: The special issue that this article seeks to engage with is very important and the issues the authors seek to contribute to are very important and timely. However, in its current structure, the article does not make such a contribution. It reads as either lists of findings (in bullet points) or short answers to questions. The discussion at the end addresses the research questions listed earlier and answers them in bullet points. A more solid and causality based narrative us necessary.

Response 2: We extended the findings and the discussion in a narrative way.

Point 3: The analysis of the data is also too descriptive. It references the quantity and topics in the articles analyzed rather than reflecting on the relevance of the literature for the overall topic of accounting and accountability.

Response 3: We have expanded and improved both the quantitative and the qualitative analysis, reflecting on the relevance of the literature for the accounting scholarship. Moreover, we have expanded the qualitative analysis, adding a more deep analysis of the contents of the papers falling within the sample, with special reference to accounting and reporting issues.

Point 4: There is no clear connection to the climate change regime and to sustainable development.

Response 4: Following your suggestions, we have added this explanation: According to Doni (2020) the connection to the climate change regime and the sustainable development is clear: the renewable energy can improve the energy access and health benefits due to the reduction of polluting emissions; measures on climate change can also improve social issues such as fragility, displacement, migration and conflict. Furthermore, climate change allows to obtain a successful implementation of all 17 SDGs.

Point 5: The article is about research agendas in accounting, and it is not clear whether this fits the purpose of the special issue for which this article is intended. Based on the title of the article, and the overall topic of the special issue “Accounting and Accountability for the SDGs”, I was expecting an analysis of what is being done to generate information and accountability that connects efforts on climate change to the indicators in SDG13.

Response 5: Following your suggestions, we have added this explanation in the conclusions “The impact of climate change accounting on SDG 13 targets and indicators is strong. Specifically, the analysis shows that the link arises in terms of information and data that accounting provides (CO2 and greenhouse gas emissions) to implement firm's adaptation and mitigation strategies: in fact the main variables that can be subjected to managerial control are the CO2 and greenhouse gas emissions.”

Point 6: The article establishes the idea of accounting, accountability and reporting, but it is only addresses the first issue. Yet, from the perspective of the SDGs the other two are even more relevant.

Response 6: Thank you for your comment. We have corrected in the whole paper this mistake. We explained (in § 1) that the paper addresses the special issue topic of “Accounting and Accountability for SDGs”, focusing upon two particular aspects within this topic, i.e. accounting and reporting.

Point 7: The structure of the abstract does not explain the importance of the topic.

Response 7: Thank you for your comment. The abstract now starts with an explanation of the importance of sustainability for corporations, especially in the last years. Many stakeholders expect companies to implement sustainability-oriented practices and report on these actions and their results. As a consequence, corporate accountability and, more specifically, corporate accounting and reporting, should focus not only on financial, social and environmental performance, but also on sustainability-related aspects. Among these aspects, climate change is becoming increasingly important for companies, which must take action to counter the effects of their activities on climate change and inform their stakeholders about these actions and their effects. See rows 16-23.

Point 8: The article refers to different categories of reporting related to sustainable development but those are not defined or explained. And this makes it more difficult to understand the discussion.

Response 8: Thank you for your comment. In § 4.5 (Qualitative analysis) we explained on which categories of accounting and reporting related to sustainable development the papers within the sample focus on.

Point 9: The authors state that sustainable development (as a concept) is the main driver for reporting. This is conflation of concepts and inaccurate.

Response 9: Thank you for your comment. We deleted this sentence.

Point 10: There are a number of issues with the methodology. The bibliometrics approach is not explained.

Response 10: Thank you for your comment. § 3 (Materials and Methods) has been restructured and partly rewritten. We described in more detail the research method, synthesizing the main phases into which the review has been structured. Thereafter, we explained each of these phases into a specific subsection. In this way, we also explained the bibliometric approach. Please see the new sections 3.1 and 3.2, that focus on the bibliometric approach.

Point 11: There are a number of issues with the methodology. [...] and there is no connection between this type of analysis and the SDGs.

Response 11: Thank you for your comment. We adopted a bibliometric approach in order to review the literature about accounting/reporting and the SDG n. 13. Our aim was not to analyze the relationship between accounting/reporting and all SDGs, but only with the specific SDG related to climate change.

Point 12: Many of the graphs are very difficult to understand. What is their value added?

Response 12: Thank you for your comment. We deleted some figures, leaving only the most significant for the analysis. We also expanded the explanation of the remaining figures.

Point 13: It is difficult to follow the relation from the data presented and the conclusions.

Response 13: Thank you for your comment. We re-wrote part of the conclusions to better explain the relationship between the data and the conclusions.

Point 14: The categories on reporting presented on the discussion around the (C) variable were not explained at any point. It makes it difficult to understand the meaning of the results.

Response 14: Thank you for your comment. We explained what are the main types of reporting addressed by the papers within the sample.

Point 15: The paper does not offer a clear contribution to the core issue of the SDGs and fails to explain why an expanded research agenda on climate change accounting would be important for sustainability.

Response 15: Thank you for your comment. We explained why accounting research on climate change would be important for sustainability.

Reviewer 3 Report

Climate change accounting and reporting: A systematic literature review

In my opinion, this paper is a good piece of research and a relevant contribution to the literature. It is clear that a great deal of effort went into the preparation of this manuscript, and I commend the authors on their progress so far. However, I think it has some minor suggestions that should be improved by the authors. So my recommendation is minor revisions. I detail below the minor problems.

In general, I highlight the accuracy of the paper; it shows an adequate and relevant methodology, statistics and observation, the conclusions refer to evidence of the paper; moreover, the paper presents comprehensive references and a clear presentation using useful tables that support the relevant evidence of the paper. I did not find mark typos and grammatical errors. Nonetheless, I have a few major and minor points that I consider the authors) should address.

  • Please, authors must detail the purpose of their study more precisely in the first lines of the abstract and provide more information about their evidence. Moreover, please, try to synthetize the abstract. 
  • The introduction section is concise and informative. Nonetheless, please, reinforce and expand the contributions of this paper to previous literature, referring to the gap of previous studies that your paper covers. Please, consider that contributions summarize the main relevance of your paper and they are necessary for readers in order to know and understand the gap in literature that the paper solves.
  • I consider that section 2 should be rewritten in order for there to be a greater connection between paragraphs. In the current version, on numerous occasions, there are one-sentence paragraphs. Check it out, please.
  • In the comment of results, please, reinforce the discussion of the evidence here obtained with respect to previous studies
  • Conclusions: Improve the practical and theoretical implications of this research.
  • Authors should make an effort to standardize the presentation of the tables and their format (please adapt them to the word format, since they appear as images).
  • Consider the length of the paper. They should try to reduce the length of it.
  • Please back to the sustainability temple and correct the contribution part. Using the whole name is not correct.

Finally, I would like to remark that the paper is potentially publishable and the results of this investigation could be of interest for the researches of this area.

Author Response

Point 1: Please, authors must detail the purpose of their study more precisely in the first lines of the abstract and provide more information about their evidence. Moreover, please, try to synthesize the abstract.

Response 1: Thank you for the suggestions. We detailed the purpose of the study at the beginning of the abstract (rows 23-25) and synthesized it.

Point 2: The introduction section is concise and informative. Nonetheless, please, reinforce and expand the contributions of this paper to previous literature, referring to the gap of previous studies that your paper covers. Please, consider that contributions summarize the main relevance of your paper and they are necessary for readers in order to know and understand the gap in literature that the paper solves.

Response 2: Thank you for your suggestion. We referred to the gap of previous studies in § 2, reinforcing and expanding the contribution of this paper to previous literature.

Point 3: I consider that section 2 should be rewritten in order for there to be a greater connection between paragraphs. In the current version, on numerous occasions, there are one-sentence paragraphs. Check it out, please.

Response 3: Thank you for your suggestion. As stated before, we modified § 2 in order to have a greater connection between paragraphs. Moreover, we eliminated the one-sentence paragraphs.

Point 4: In the comment of results, please, reinforce the discussion of the evidence here obtained with respect to previous studies.

Response 4: Thank you for your suggestion. We reinforced the evidence obtained with respect to previous studies in the conclusions in the last sections of the paper.

Point 5: Conclusions: Improve the practical and theoretical implications of this research.

Response 5: Thank you for your suggestion. We emphasized the practical and theoretical implications of this research in the conclusions.

Point 6: Authors should make an effort to standardize the presentation of the tables and their format (please adapt them to the word format, since they appear as images).

Response 6: Thank you for your comment. We eliminated some tables and figures and standardized the remaining tables and figures. The figures were downloaded from the bibliometrix software, which does not allow saving in Word format. However, we have improved the graphic aspect and the readability of the figures.

Point 7: Consider the length of the paper. They should try to reduce the length of it.

Response 7: Thank you for your comment. We eliminated some tables, figures, and textual parts of the paper. In order to answer other reviewers’ comments, we also added new parts.

Point 8: Please back to the sustainability temple and correct the contribution part. Using the whole name is not correct.

Response 8: Sorry for the mistake. We corrected the author contributions section.

Round 2

Reviewer 2 Report

The article has definitively improved and the structure is much better. The discussion is much more detailed and putting the main points in a narrative rather than in bullet points makes a difference.   However, the connection to sustainable development is not entirely clear. With that in mind, I will suggest the following:   - At the beginning, the authors reflect on how the accounting literature has reflected different SDGs. They mention different references for each SDG but it would be good to present these more as examples, saying how they are connecting accounting to sustainable development. - The concept of reporting should be further developed. Or at least explained conceptually. Reporting to governments or reporting in terms of corporate responsibilities and commitments. - Some of the findings need to clarify that this is about the research on accounting and its relation to climate change, not about the actual practices among corporations and countries. - I find difficult the relations between what the authors are mentioning that accounting and reporting for climate change should measure (CO2 and Greenhouse emissions) and SDG13. The specific details of the goal offer no direct relation, so it will be difficult to calculate that in terms of the contribution to the specific SDG. This is not a problem, but the article needs to be clear about the connection to sustainable development. And also about the role that corporations and firms have in the overall national efforts for adaptation and mitigation.   There are also a couple of issues in edits that need attention. Line 1665 needs to capitalize GHG and around line 2615 there are several “climate changes” that should read “climate change”.  

Author Response

Point 1: At the beginning, the authors reflect on how the accounting literature has reflected different SDGs. They mention different references for each SDG but it would be good to present these more as examples, saying how they are connecting accounting to sustainable development.

Response 1: Thank you for the suggestions. We inserted a brief explanation of some of the cited sources, focusing on how they connect accounting/reporting to sustainable development. The explanation focuses only on the sources whose link between accounting/reporting and sustainable development seemed us to be more relevant for our analysis. Here we copy what we wrote to address your suggestion:

“Some authors started dealing with the relationship between accounting and SDGs [10; 18; 50], focusing on:

  • SDG n. 6 (clean water and sanitation) [e.g. 33;48]. [33] focuses on the intersection between accounting and human rights. Within the latter, the author questions if the access to information can be considered as a human right, focusing specifically on environmental information related to water sources. [48] focuses on the reporting of sustainability information to public water companies’ stakeholders.
  • SDGs n. 5 (gender equality), SDG n. 10 (reduced inequalities), and 16 (peace, justice and strong institutions) [e.g. 4; 56; 57; 77]. Among this literature, it is worth mentioning [56], containing an introductory analysis of the potential role of the International Accounting Standards Board (IASB) for the application of relevant human rights norms, and [77], dealing with social accounting research related to economic inequality, and calling for a deep involvement of social accountants in public debates about the future of resource distribution.
  • SDGs n. 14 and n. 15 (life below the water/life on land) [e.g. 8; 27; 44; 72; 78]. Among this literature, it is worh mentioning [27], dealing with the potential role of environmental accounting in the reconstruction of the knowledge of the social, economic, and environmental risks of salmon farming and in the choice of adopting organic production methods. [72] extends the research stream of biodiversity accounting, trying to verify if Jones’ natural inventory model is applicable to a specific local context.”

Point 2: The concept of reporting should be further developed. Or at least explained conceptually. Reporting to governments or reporting in terms of corporate responsibilities and commitments.

Response 2: Thank you for your suggestion. We developed the concept of reporting in § 2, focusing also on the main stakeholders pressures behind climate change disclosure. Here we copy what we wrote to address your suggestion:

“In recent decades, public and private organizations are expected to be more and more accountable for the impact of their actions on the environment. Stakeholders expect, in particular, information on the impact of organizations’ actions on climate change [65; 1]. In this context, there was an evolution of sustainability and climate change-related information provided by public and private organizations, increased especially since the issue of the Kyoto protocol in 1997.

In the 1980s and 1990s, environmental information was mainly provided in the annual reports of companies operating in different countries [65]. Thereafter, environmental information was usually provided through voluntary information displayed in sustainability reports published on the organizations’ websites. This information often incorporated GRI indicators as well as other economic, environmental and social features [65]. As a consequence, sustainability reporting has gained relevance as a means to be accountable for the environmental impacts of organizations’ activity. The need for high-quality reporting about sustainable development, SDGs, and financial balance was highlighted during the 32nd session of the Intergovernmental Working Group of Experts in International Standards of Accounting and Reporting, held in Geneva in 2015 [54]. Accounting practitioners and scholars play a relevant role in the achievement of the SDGs (and of climate change-related objectives), since they are involved in accounting, reporting, and auditing activities related to these goals.

The disclosure of climate change-related information falls within a “grey area” between carbon/GHG financial accounting and non-financial disclosure (the latter is also known as “narrative disclosure”) [52]. In this context, academic research about climate change is focusing both on sustainability reporting, which has now become a recognized research area in accounting studies [74], and on the integration of climate change reporting into financial reports [52; 14].

According to Haque and Islam [32], the pressures behind climate change accountability and disclosure come from various stakeholders, i.e. government bodies (regulators), institutional investors, environmental NGOs, and media accounting professionals.

Reporting to government bodies (i.e. regulators) is mainly aimed at fulfilling obligations connected to the legislation on emissions rights (tradable emission rights and obligations related to emission trading schemes). Reporting in terms of corporate responsibilities and commitments is mainly aimed at supporting the image and reputation of the organization towards various types of stakeholders.

According to Solomon et al. [74], among the stakeholder pressures, climate change reporting is mainly driven by risk and risk management motivation. In other words, the need to disclose climate change information is induced by institutional investors’ (who are the main financial stakeholders of corporations) belief that climate change: (a) is a relevant risk; (b) is the most important issue within sustainability; (c) is a relevant aspect for organizations’ clients, who need that climate change-related risks are managed in their portfolio investment.”

Point 3: Some of the findings need to clarify that this is about the research on accounting and its relation to climate change, not about the actual practices among corporations and countries.

Response 3: Thank you for your suggestion. We clarified in the conclusions that the findings relate to the research on accounting and climate change. Here we copy what we wrote to address your suggestion:

“According to Doni [20], the connection between the climate change regime and sustainable development is clear: renewable energy can improve energy access and health benefits related to the reduction of polluting emissions; actions against climate change can also improve social issues such as fragility, displacement, migration and conflict. Furthermore, climate change allows obtaining a successful implementation of all 17 SDGs. The impact of climate change accounting on SDG no. 13 targets and indicators is strong. Specifically, this literature review shows that the link arises in terms of information and data that accounting provides (CO2 and greenhouse gas emissions) to implement the adaptation and mitigation strategies of firms: in fact, the main variables that can be subjected to managerial control are the CO2 and greenhouse gas emissions.”

Point 4: I find difficult the relations between what the authors are mentioning that accounting and reporting for climate change should measure (CO2 and Greenhouse emissions) and SDG13. The specific details of the goal offer no direct relation, so it will be difficult to calculate that in terms of the contribution to the specific SDG. This is not a problem, but the article needs to be clear about the connection to sustainable development. And also about the role that corporations and firms have in the overall national efforts for adaptation and mitigation.

Response 4: Thank you for your suggestion. In § 1 we explained that, “According to SDG n. 13, climate change is induced by global warming, which is strongly related to rising CO2 and GHG emissions. To limit climate change, it is important to act on its causes.

The interest of accounting scholars towards anthropogenic induced global climate change reduction has therefore forceful links with GHG emissions accounting and reporting. The latter is a broad set of disclosures dealing with the impact of human and corporate activities on the climate, e.g. carbon emissions disclosures, other greenhouse gas (GHG) emissions disclosures, and footprint-related disclosures. The environmental issue is relevant not only for governments, but also for public and private organizations from different activity sectors (Prado Lorenzo et al., 2009). Companies (especially large ones) are responsible for great quantities of polluting emission into the atmosphere. It is reasonable to expect them to be widely involved in the reduction of such emissions, in the reporting of the activities carried out to achieve this goal, and in the results actually achieved. Consistent with these expectations, the papers analysed in this literature mainly focused on accounting and reporting for CO2 and GHG emissions.”

Point 5: There are also a couple of issues in edits that need attention. Line 1665 needs to capitalize GHG and around line 2615 there are several “climate changes” that should read “climate change”.

Response 5: Thank you for your suggestion. We corrected “ghg” into “GHG” and “climate changes” into “climate change”.

Reviewer 3 Report

Congrats for the revised manuscript. The authors have solved all my concerns. In my opinion, the paper can be accepted as is.

Author Response

Thank you very much for your suggestions. They helped us in the improvement of the paper.